# Odors drive feeding through gustatory receptor neurons in *Drosophila*

Hongping Wei[1], Thomas Ka Chung Lam[2], Hokto Kazama[1,3,4]*

[1]RIKEN Center for Brain Science, Saitama, Japan; [2]Neuroengineering Laboratory, Brain Mind Institute & Interfaculty Institute of Bioengineering, EPFL, Lausanne, Switzerland; [3]RIKEN CBS-KAO Collaboration Center, Saitama, Japan; [4]Graduate School of Arts and Sciences, The University of Tokyo, Tokyo, Japan

## eLife Assessment

This study **convincingly** demonstrates that odors evoke a feeding response in *Drosophila*, mediated by gustatory receptors and observed as a proboscis extension. The evidence is comprehensive, encompassing behavior, functional imaging and electrophysiology. This **important** results on the molecular and cellular basis of multimodal integration across olfaction and gustation will be of interest for the study of chemosensation, sensory biology, and animal behavior.

*For correspondence:
hokto.kazama@riken.jp

Competing interest: The authors declare that no competing interests exist.

## Abstract

Odors are intimately tied to the taste system to aid food selection and determine the sensory experience of food. However, how smell and taste are integrated in the nervous system to drive feeding remains elusive. We show in *Drosophila* that odors alone activate gustatory receptor neurons (GRNs), trigger proboscis extension reflex (PER), a canonical taste-evoked feeding behavior, and enhance food intake. Odor-evoked PER requires the function of sugar-sensing GRNs but not olfactory organs. Calcium imaging and electrophysiological recording show that GRNs directly respond to odors. Odor-evoked PER is mediated by the Gr5a receptor, and is bidirectionally modulated by olfactory binding proteins. Finally, odors and sucrose co-applied to GRNs synergistically enhance PER and food consumption. These results reveal a cell-intrinsic mechanism for odor-taste multimodal integration that takes place as early as in GRNs, indicating that unified chemosensory experience is a product of layered integration in peripheral neurons and in the brain.

## Introduction

Feeding behavior is an outcome of intricate interactions among multiple external and internal senses (*Boesveldt and de Graaf, 2017*; *McCrickerd and Forde, 2016*; *Spence, 2015*). Of particular relevance is an interplay between the senses of smell and taste. Olfactory and gustatory systems prime feeding by signaling food presence, guiding food choice, and inducing anticipatory responses in digestive organs (*Boesveldt and de Graaf, 2017*; *McCrickerd and Forde, 2016*). During consumption, they synergistically define the sensory experience of food together with other cues (*Spence, 2015*). Although imaging studies have identified neural correlates of odor-taste interaction in the brain (*Small, 2012*), a mechanistic understanding of chemosensory integration is still lacking.

*Drosophila* offers an attractive model to address this issue, where peripheral chemosensory systems are well-characterized (*Scott, 2018*; *Wilson, 2013*), and feeding behavior can be quantified by measuring the strength of PER (*Dethier, 1976*) and the amount of subsequent food consumption. PER is an initial step of feeding evoked by the presentation of sugar to the proboscis or legs housing GRNs. Previous studies reported that odors increase the rate of PER (*Oh et al., 2021*; *Reisenman and Scott, 2019*; *Shiraiwa, 2008*) as well as food intake (*Reisenman and Scott, 2019*), exemplifying

olfactory enhancement of feeding in *Drosophila*. However, aside from the involvement of olfactory receptor neurons (*Oh et al., 2021*; *Shiraiwa, 2008*), biological mechanisms underlying this multisensory enhancement remain elusive.

Here, we investigated the impact of odors on feeding behavior and found direct contribution of GRNs to odor-evoked PER, odor-taste integration, and odor-driven enhancement of food consumption, indicating that multisensory integration commences at the very periphery.

## Results

### Odors alone evoke PER

To examine how odors affect PER, we built a behavioral recording setup in which a tastant and multiple odors with diverse innate values (*Badel et al., 2016*) can be applied to individual, tethered flies (*Figure 1A*). To quantify the movement of proboscis, we used a deep learning-based, markerless pose estimation algorithm (*Mathis et al., 2018*) and tracked the trajectory of three segments constituting the proboscis. The angles made between the segments (rostrum, haustellum, and labellum angles) were subsequently calculated over time to detect PER (*Figure 1A–C*; see Methods for the definition of PER). The results of this quantification and manual scoring of PER were comparable (*Figure 1—figure supplement 1C*). Unexpectedly, we found that the odors alone evoked repetitive PER without an application of a tastant (*Figure 1D–G*, and *Figure 1—video 1*). Different odors evoked PER with different probability (*Figure 1E*), latency (*Figure 1—figure supplement 1A*), and duration (*Figure 1F and G*, *Figure 1—figure supplement 2*). Stimulus-triggered PER was not observed in response to solvents or air (*Figure 1D–G*). Odors evoked PER even at the concentration of $10^{-2}$, which induces stimulus-specific activity in the olfactory pathway (*Badel et al., 2016*; *Endo et al., 2020*; *Kato et al., 2023*). The level of PER did not correlate with the innate value of odors, previously determined by quantifying the odor preference of flies in a behavioral arena (*Badel et al., 2016*; Pearson's $R$=0.33, p=0.52). Odor-evoked PER was observed in other genetic backgrounds and another *Drosophila* species as well, albeit at a lower level with different tuning (*Figure 1—figure supplement 3*).

Odor-evoked PER resembled taste-evoked PER in multiple aspects. First, its trajectory was similar to that of taste-evoked PER (*Figure 1—figure supplement 1B*, *Figure 1—video 1*) and was consistent with the reported movement of proboscis during sucrose-evoked PER (*Schwarz et al., 2017*). The trajectories were indistinguishable between the tested odors (*Figure 1—figure supplement 1B*). Second, it showed concentration dependency (*Dethier, 1976*; *Wang et al., 2004*). The probability and duration of PER increased as a function of stimulus concentration (*Figure 1E and G*). Third, it depended on the metabolic state just as taste-evoked PER (*Inagaki et al., 2014*). The duration of PER was higher in starved as compared to fed flies (*Figure 2—figure supplement 1*).

In sum, odors initiate feeding behavior in *Drosophila*.

### Odors evoke PER through GRNs

To confirm that this behavior is mediated by the olfactory system, we repeated the experiment after removing the sensory organs housing the olfactory receptor neurons, namely the antennae and the maxillary palps. However, this manipulation only attenuated and did not eliminate odor-evoked PER (*Figure 2A*), suggesting that the olfactory system modulates but is not required for odor-evoked PER.

We thus turned to another chemosensory system, the gustatory system. To examine its involvement in odor-evoked PER, we expressed Kir2.1 (*Paradis et al., 2001*) in different types of GRNs using type-specific Gal4 drivers, each of which mediates a specific taste modality (*Scott, 2018*). Gr5a GRNs detect sugar and mediate food acceptance behaviors, including PER (*Dahanukar et al., 2001*; *Wang et al., 2004*). Suppression of Gr5a GRNs significantly reduced odor-evoked PER, indicating that sweet-sensing neurons detect odors and drive PER (*Figure 2B and C*). Bitter-sensing Gr66a GRNs, on the other hand, have been shown to inhibit sweet-evoked PER (*Wang et al., 2004*). However, expression of Kir2.1 in these neurons had little effect (*Figure 2B and D*). We hypothesized that this is due to our use of starved flies because Gr66a pathway is reported to be downregulated in a starved condition (*Devineni et al., 2019*; *Inagaki et al., 2014*). Indeed, flies of the same genotype showed increased PER to odors in a fed state (*Figure 2G*), indicating that Gr66a GRNs counteract PER. Two other tastes, water and low salt can also induce PER (*Cameron et al., 2010*; *Wang et al., 2004*), which are detected by Ppk28 and Ir94e GRNs, respectively (*Cameron et al., 2010*; *Jaeger et al.,*

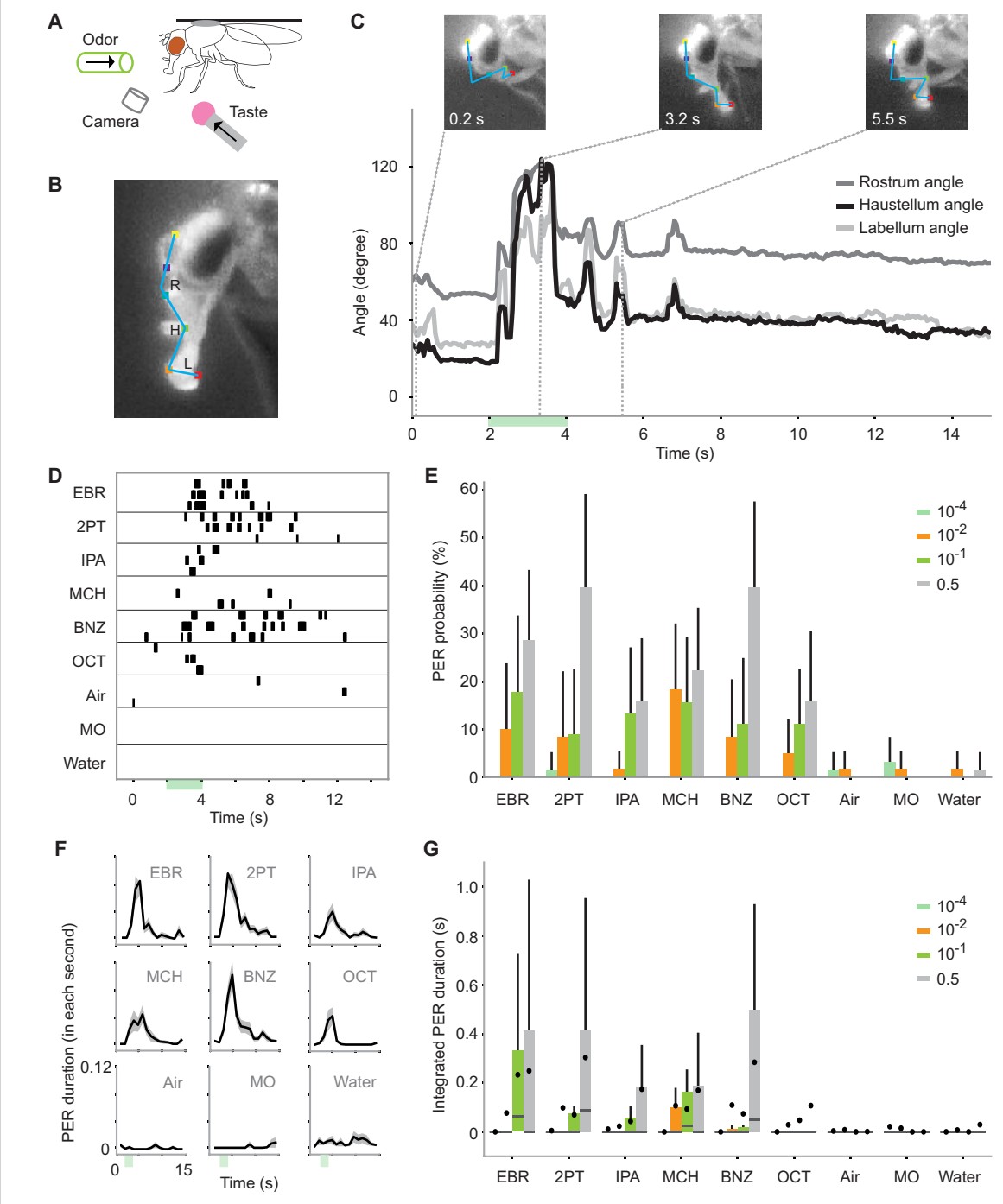

**Figure 1.** Odors alone evoke proboscis extension reflex (PER). (**A**) Schematic for recording odor-evoked PER. (**B**) Six points on the head are tracked by DeepLabCut. The movement of the proboscis was characterized by the positions of three segments of the proboscis and the angles between the segments, namely the rostrum angle (R), haustellum angle (H), and labellum angle (L). (**C**) Example odor-evoked PER. Top, images of a proboscis in a retracted (0.2 s), fully extended (3.1 s), and partially extended (5.5 s) state. Bottom, the three angles over time. Green bar indicates an odor application period. Odor is ethyl butyrate. (**D**) PER to nine odors in an example fly. Each row corresponds to a trial and each tick mark indicates the timing of PER. The odors used are as follows: ethyl butyrate (EBR), 2-pentanone (2PT), isopentyl acetate (IPA), 4-methylcyclohexanol (MCH), benzaldehyde (BNZ), 3-octanol (OCT), and mineral oil (MO). (**E**) PER probability in wild-type flies in response to different concentrations of odors. The same stimulus was used for air, mineral oil, and water controls. n=21, 26, 27, and 28 flies for $10^{-4}$, $10^{-2}$, $10^{-1}$, and 0.5 concentration groups, respectively. Error bar, standard error of the mean. PER probability was different between odor concentrations and identities (p=3.0e-05 and 2.0e-7 for odor concentration and identity factors, Scheirer–Ray–Hare test). (**F**) PER duration in wild-type flies. The black lines indicate the mean and the shaded areas indicate the standard error of the mean. n=28 flies. (**G**) Integrated PER duration in wild-type flies in response to different concentrations of odors. The data is from the same flies as in

*Figure 1 continued*

E. Integrated PER duration was different between odor concentrations and identities (p=3.6e-13 and 3.5e-11 for odor concentration and odor identity factors, Scheirer–Ray–Hare test). Box plots indicate the median (gray line), mean (black dot), quartiles (box), and 5–95% range (bar).

The online version of this article includes the following video and figure supplement(s) for figure 1:

**Figure supplement 1.** Characteristics of proboscis extension reflex (PER) in wild-type flies.

**Figure supplement 2.** Proboscis extension reflex (PER) evoked by additional odors.

**Figure supplement 3.** Odor-evoked proboscis extension reflex (PER) in flies with different genetic backgrounds and in another species.

**Figure 1—video 1.** Odor-evoked proboscis extension reflex (PER).

https://elifesciences.org/articles/101440/figures#fig1video1

*2018*). However, suppression of these cell types did not affect odor-evoked PER (*Figure 2B, E and F*). Together, Gr5a GRNs enhance, whereas Gr66a GRNs inhibit odor-evoked PER.

As GRNs are housed in multiple external organs, including the labella at the tip of proboscis, the legs, the wing margins, and the ovipositor (*Scott, 2018*), we next sought to narrow down the GRNs responsible for odor-evoked PER. We found that removal of the legs and wings did not abolish the behavior (*Figure 2—figure supplement 2*). Moreover, activation of the ovipositor induces egg laying rather than PER (*Dethier, 1976*). These results suggest that GRNs act predominantly in the labella to induce odor-evoked PER.

## Gustatory receptors and olfactory binding proteins mediate odor-evoked PER

We have shown that Gr5a GRNs tuned to tastes trigger PER in response to a wide range of odors. Because GRNs do not express olfactory receptors (*Davie et al., 2018*; *Li et al., 2021*; *Scott et al., 2001*), either gustatory receptors or some other molecules are likely to interact with odorants (*Jones et al., 2007*; *Kwon et al., 2007*). Gr5a receptor is a primary candidate as Gr5a GRNs drive odor-evoked PER (*Figure 2B and C*). Consistent with this hypothesis, odor-evoked PER was nearly undetectable in the Gr5a mutant but spared in the control (*Figure 2H*). This demonstrates that Gr5a receptor is necessary for odor-evoked PER.

Odorant binding proteins (OBPs) are another set of molecules that could mediate odor detection by GRNs. They are thought to bind and help transport hydrophobic odorants to chemosensory receptors through aqueous lymph (*Larter et al., 2016*). OBPs are expressed in taste as well as olfactory sensilla (*Galindo and Smith, 2001*) and modulate sugar-evoked PER (*Swarup et al., 2014*), raising the possibility that those in the taste sensilla also mediate odorant detection. To test this, we conducted a genetic screen by expressing individual Obp RNAi constructs with *tubulin-Gal4*. We targeted 9 Obp genes that are expressed highly in the labella (*Cameron et al., 2010*) but in trace amounts in the antenna (*Larter et al., 2016*) to examine the function of OBPs in the taste sensilla. Knockdown of *Obp18a*, *Obp57d/e*, and *Obp57e* genes decreased, whereas that of *Obp49a* gene increased odor-evoked PER (*Figure 2I and J*, *Figure 2—figure supplement 3*). Knockdown of *Obp19b*, *Obp56g*, *Obp56h*, *Obp57a/c*, and *Obp83c* genes did not have a significant effect on odor-evoked PER (*Figure 2K*, *Figure 2—figure supplement 3*). The increased PER following *Obp49a* knockdown likely reflects disinhibition of sweet-sensing GRNs as Obp49a mediates the inhibitory impact of bitter chemicals on the activity of sweet-sensing GRNs (*Jeong et al., 2013*). These results suggest that OBPs modulate odor detection by GRNs.

## GRNs directly respond to odors

To gain direct evidence that GRNs respond to odors, we performed two-photon calcium imaging and single-sensillum electrophysiological recording that have complementary properties (*Figures 3 and 4*). Calcium imaging of GRN axons using a cell-specific driver reports the population activity of neurons but can unambiguously ascribe the signals to genetically identified GRNs. On the other hand, while single-sensillum electrophysiology cannot uniquely ascribe the signals to genetically defined GRNs, it is sensitive and fast enough to detect individual spikes.

For imaging, we expressed a calcium indicator GCaMP6s (*Chen et al., 2013*) in Gr5a, Gr66a, Ppk28, or Ir94e GRNs and scanned the axons in the subesophageal zone of the brain (*Figure 3A*). Consistent with the contribution of Gr5a and Gr66a GRNs to odor-evoked PER, robust odor responses

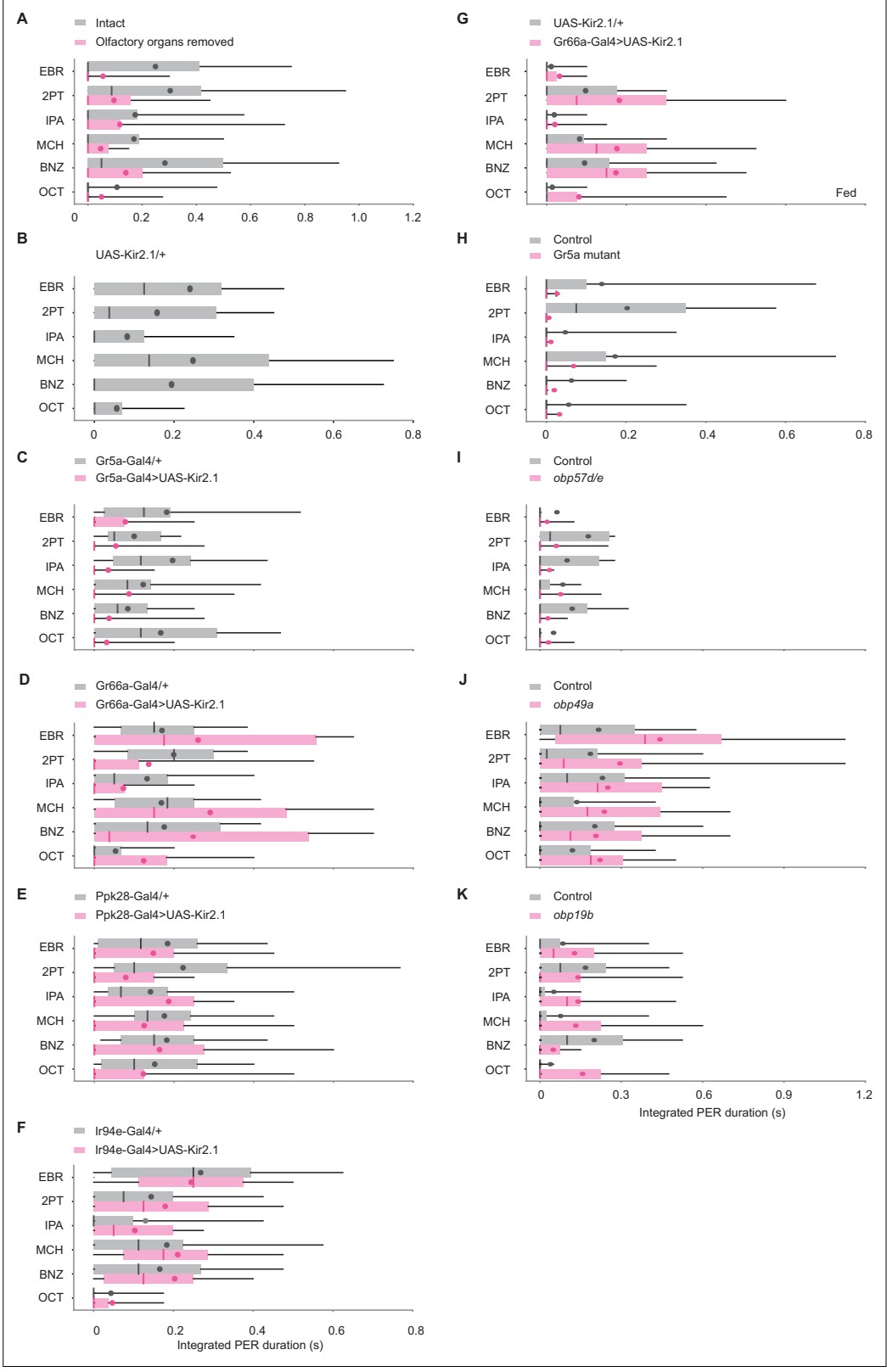

**Figure 2.** Odors evoke proboscis extension reflex (PER) through gustatory receptor neurons (GRNs). (**A**) Integrated PER duration in intact (n=28) and olfactory organs removed (n=32) wild-type flies. Flies with olfactory organs removed show less PER compared to control flies (p=0.0079 and 0.10 for group and odor factors, Scheirer–Ray–Hare test). (**B–F**) Integrated PER duration in UAS control UAS-Kir2.1/+, n=20, (**B**), GRN silenced (Gr5a-Gal4>UAS-

*Figure 2 continued on next page*

*Figure 2 continued*

Kir2.1, n=25 for C; Gr66a-Gal4>UAS-Kir2.1, n=24 for D; Ppk28-Gal4>UAS-Kir2.1, n=28 for E; Ir94e-Gal4>UAS-Kir2.1, n=23 for F) and Gal4 control flies (Gr5a-Gal4/+, n=31 for C; Gr66a-Gal4/+, n=21 for D; Ppk28-Gal4/+, n=28 for E; Ir94e-Gal4/+, n=30 for F). Each silenced line was compared to the UAS control in B (p=4.9e-5, 0.49, 0.17, 0.11, and 0.12, 0.0014, 0.55, 1.2e-5 for genotype and odor factors for C-F, Scheirer–Ray–Hare test). Data were also compared between each silenced line and its Gal4 control (p=1.0e-15, 0.28, 0.00062, 0.14, and 0.71, 0.0012, 0.56, 5.6e-9 for genotype and odor factors for C-F, Scheirer–Ray–Hare test). (G) Integrated PER duration in control (UAS-Kir2.1/+, n=21) and Gr66a GRN-silenced (Gr66a-Gal4>UAS-Kir2.1, n=21) flies in a fed state. Silencing of Gr66a GRNs enhances odor-evoked PER in a fed state (p=1.8e-4 and 0.041 for genotype and odor factors, Scheirer–Ray–Hare test). Odor concentration was $10^{-1}$. (H) Integrated PER duration in control (n=29) and Gr5a mutant (n=29) flies. PER is severely reduced in mutant flies (p=1.1e-4 and 0.32 for genotype and odor factors, Scheirer–Ray–Hare test). Control: *EP(x)496* flies. Mutant: *ΔEP(x)–5* flies. (I–K) Integrated PER duration in control and Obp RNAi flies. Obp57d/e RNAi flies show reduced (I, n=33 and 25 for control and RNAi, p=0.039 and 0.48 for genotype and odor factors, Scheirer–Ray–Hare test), Obp49a RNAi flies show enhanced (J, n=24 and 20 for control and RNAi, p=0.0066 and 0.16 for genotype and odor factors, Scheirer–Ray–Hare test), and Obp19b RNAi flies show similar levels of PER (K, n=24 and 21 for control and RNAi, p=0.24 and 0.26 for genotype and odor factors, Scheirer–Ray–Hare test) as compared to their respective controls, which are *tubulin-Gal4/y,w^{1118};P{attP,y[+],w[3']}* (for I), *tubulin-Gal4/w^{1118};P{VDRCsh60200}attP40* (for J), and *tubulin-Gal4/w^{1118}* (for K). Box plots indicate the median (gray and red lines), mean (black and red dots), quartiles (box), and 5–95% range (bar).

The online version of this article includes the following figure supplement(s) for figure 2:

**Figure supplement 1.** Dependence of odor-evoked proboscis extension reflex (PER) on a feeding state.

**Figure supplement 2.** Wings and legs are not required for odor-evoked PER.

**Figure supplement 3.** Contributions of odorant binding proteins (OBPs) to odor-evoked proboscis extension reflex (PER).

were observed in these neurons. They responded to all the tested odors but not to control stimuli, with distinct tuning (*Figure 3B–F*, *Figure 3—figure supplement 1A–F* and *Figure 3—figure supplement 2*), in a concentration-dependent manner (*Figure 3—figure supplement 1G and H*, *Figure 3—figure supplement 3B*). A linear combination of Gr5a and Gr66a GRN responses well predicted the magnitude of odor-evoked PER with the former and the latter contributing to enhancement and suppression of PER, respectively (*Figure 3G*). Odor and sucrose responses were positively correlated in Gr5a GRNs (*Figure 3—figure supplement 1I*). By contrast, Ppk28 and Ir94e GRNs did not respond to any odors (*Figure 3—figure supplement 1J and K*), matching their little contribution to odor-evoked PER (*Figure 2B, F and G*).

These responses could reflect the activity originating in GRNs or that from a central olfactory circuit impinging on GRN's presynaptic terminals. To distinguish between the two scenarios, we physically covered the maxillary palps with UV glue to block the input from the olfactory organs and applied odors (antennae were already removed to gain optical access to the subesophageal zone, see Methods). This manipulation had little effect on the odor responses (*Figure 3D and F*). On the other hand, the responses were eliminated after covering the labella (*Figure 3D and F*), demonstrating that GRNs themselves sense odors.

As we observed that Gr5a receptor was necessary for odor-evoked PER, we further asked whether it is required for Gr5a GRNs to respond to odors. We found that odor responses were absent in the Gr5a mutant but spared in control flies (*Figure 3H*). This confirms that odor responses in Gr5a GRNs depend on Gr5a receptor.

For electrophysiology, we measured odor-evoked responses from individual taste sensilla on the proboscis using tip recording, a conventional technique where a glass electrode placed on the tip of a sensillum delivers a stimulus as well as records neural responses (*Figure 4*; *Hodgson et al., 1955*). We targeted large and intermediate type (L- and I-type) taste sensilla, and recorded responses to odors and sucrose. Odors were dissolved in tricholine citrate (TCC), an electrolyte that suppresses the responses of water-sensing neurons (*Wieczorek and Wolff, 1989*). L-type sensilla housing a sweet-sensing GRN robustly responded to sucrose, but also to two tested odors, 4-methylcyclohexanol and banana odor, at a rate higher than the solvent (*Figure 4A and B*). I-type sensilla housing sweet- and bitter-sensing GRNs also responded to odors (*Figure 4A and C*). These sensilla exhibited spikes with different amplitudes that may originate in different GRNs. This is consistent with our calcium imaging results showing that both Gr5a and Gr66a GRNs respond to odors (*Figure 3*). Here, we counted all

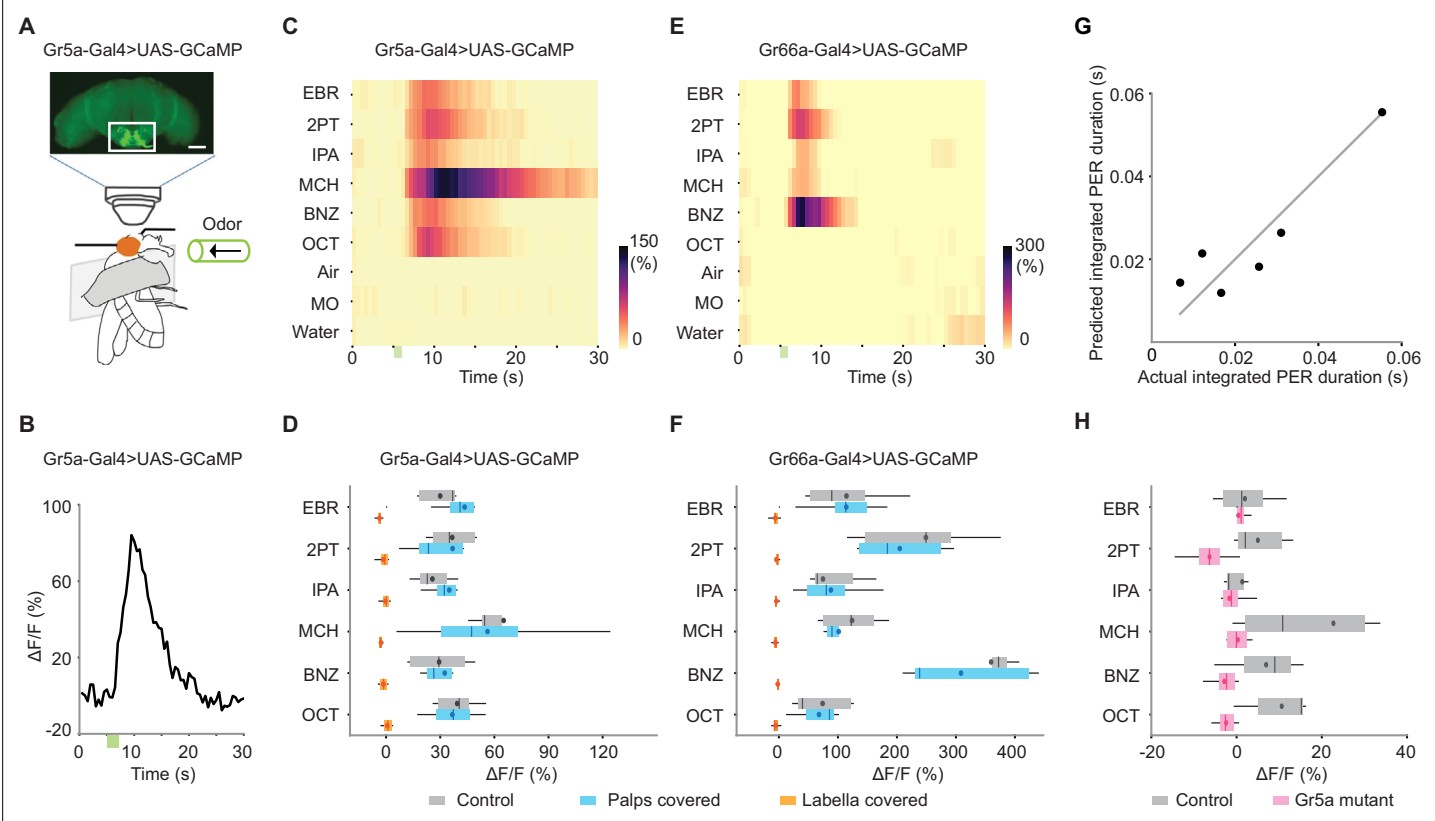

**Figure 3.** Sweet- and bitter-sensing gustatory receptor neurons (GRNs) directly respond to odors. (**A**) Schematic for two-photon calcium imaging of GRN axon termini in the subesophageal zone. The top image shows an anterior view of a brain of Gr5a-Gal4>UAS-GCaMP6s fly. White rectangle indicates the target region of calcium imaging. Scale bar: 50 μm. (**B**) Example response (ΔF/F) to ethyl butyrate. Green bar indicates an odor application period. Odor concentration was $10^{-1}$ for all the odors in all the experiments described in this figure. (**C**) Summary responses of Gr5a GRNs to nine odors. Green bar indicates an odor application period. ΔF/F of GCaMP6s fluorescence is color coded according to the scale bar (n=5). (**D**) Trial-averaged peak responses of Gr5a GRNs to individual odors. Covering the labella with glue reduced odor responses as compared to control (n=5 and 4 for control and labella covered, p=2.6e-15 and 5.1e-3 for organ condition and odor factors, mixed two-way ANOVA) but covering the maxillary palps had no effects (n=5 and 4 for control and palps covered, p=0.69 and 0.055 for organ condition and odor factors, mixed two-way ANOVA). (**E, F**) Same as in C, D, but for Gr66a GRNs. As in Gr5a GRNs, covering the labella with glue reduced odor responses as compared to control (n=6 and 5 for control and labella covered, p=3.0e-16 and 1.1e-7 for organ condition and odor factors, mixed two-way ANOVA) but covering the maxillary palps had no effects (n=6 and 5 for control and palps covered, p=0.48 and 4.2e-11 for organ condition and odor factors, mixed two-way ANOVA). (**G**) A linear combination of Gr5a and Gr66a GRN responses well predicted the magnitude of odor-evoked PER. PER was measured in Gr5a-Gal4>UAS-GCaMP flies. Each dot represents an odor. Coefficient of determination = 0.81. Weights for Gr5a and Gr66a GRNs are 5.0e-4 and −2.3e-5, indicating that the former and the latter contributes to enhancement and suppression of PER, respectively. (**H**) Odor responses are severely reduced in Gr5a GRNs in a Gr5a receptor mutant background (n=5 and 8 for control and mutant, P=8.0e-06 and 0.037 for genotype and odor factors, mixed two-way ANOVA). Box plots indicate the median (gray and red lines), mean (black and red dots), quartiles (box), and 5–95% range (bar).

The online version of this article includes the following figure supplement(s) for figure 3:

**Figure supplement 1.** Responses of GRNs to odors and sucrose.

**Figure supplement 2.** Individual responses of GRNs to odors.

**Figure supplement 3.** PER and GRN responses to various concentrations of banana odor.

the spikes and did not sort them based on their amplitude and shape, because these characteristics change depending on the spiking frequency (*Fujishiro et al., 1984*).

In sum, recording from specific types of GRNs using calcium imaging and specific types of taste sensilla using tip recording together demonstrates that GRNs in the labella sense odors in line with a recent study using a different electrophysiological technique (*Dweck and Carlson, 2023*).

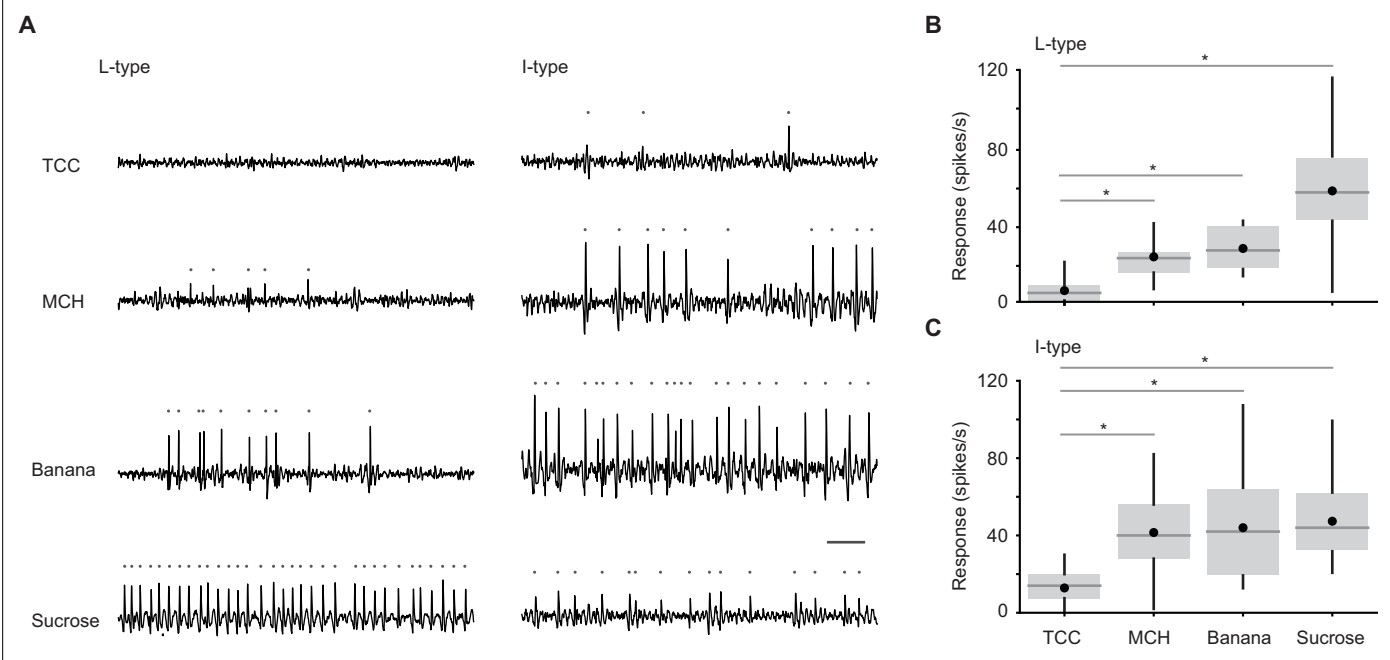

**Figure 4.** Individual taste sensilla sense odors. (**A**) Example responses of individual taste sensilla recorded using tip recording. Left, example responses to tricholine citrate (TCC) (30 mM), 4-methylcyclohexanol (MCH) (1%), banana odor (1%), and sucrose (100 mM) recorded in the same L-type sensillum; Right, example responses to the same set of stimuli in the same I-type sensillum. Traces show the activity between 200 and 700 ms after the stimulus onset. Each gray dot indicates a spike. Bar indicates 50 ms. (**B**) Average responses of L-type sensilla. (n=20, 6, 17, 20 flies for TCC, MCH, banana odor, and sucrose, respectively). Responses to stimuli are significantly larger than those to TCC (p=3.0e-18, one-way ANOVA followed by Tukey's HSD, p=0.0051, 0.0021, and 2.7e-7 for MCH, banana, and sucrose compared to TCC). (**C**) Average responses of I-type sensilla. (n=30, 9, 22, 30 flies for TCC, MCH, banana odor, and sucrose, respectively). Responses to stimuli are significantly larger than those to TCC (p=1.4e-12, one-way ANOVA followed by Tukey's HSD, p=3.9e-6, 2.0e-7, and 5.9e-12 for MCH, banana, and sucrose compared to TCC). Box plots indicate the median (gray line), mean (black dot), quartiles (box), and 5–95% range (bar). Asterisk indicates $P<0.05$.

### Odor-taste integration in single taste sensillum

We investigated whether GRNs integrate multisensory stimuli when they are presented together. We presented sucrose, banana odor, or a mixture of the two to individual taste sensilla during tip recording. We found that the responses to a mixture were stronger than those to component stimuli (*Figure 5*). While the strength of responses varied across sensilla, responses to banana odor and sucrose recorded in the same sensillum were positively correlated (*Figure 5—figure supplement 1*). These results suggest that GRNs function as an odor-taste multisensory integrator.

### GRNs integrate multisensory input to enhance PER and food consumption

We finally asked if GRNs can enhance PER and subsequent food consumption through multisensory integration. We decided to apply a mix of banana odor and sucrose as a mimic of multisensory food stimuli in the natural environment. We confirmed that the banana odor can evoke PER and GRN responses in a concentration-dependent manner (*Figure 3—figure supplement 3*). We then presented sucrose solution mixed with or without a low concentration ($10^{-4}$) of banana odor locally to the labella (*Figure 6A*). Although banana odor could not evoke PER on its own at this concentration (*Figure 3—figure supplement 3A*), we found that the addition of banana odor increased PER to sucrose, especially at low concentrations where sucrose alone can only induce unreliable PER (*Figure 6B*). Importantly, this enhancement was observed even after removing the olfactory organs (*Figure 6B*), indicating that the superadditive integration takes place in GRNs. Similar results were obtained when sucrose was mixed with different monomolecular odorants (*Figure 6C and D*).

We further examined if the amount of food consumption is also enhanced by odor-taste integration in GRNs. Using a previously reported multisensory feeding assay (*Reisenman and Scott, 2019*), we quantified the amount of sucrose consumed by freely behaving flies with or without odors (*Figure 6E*).

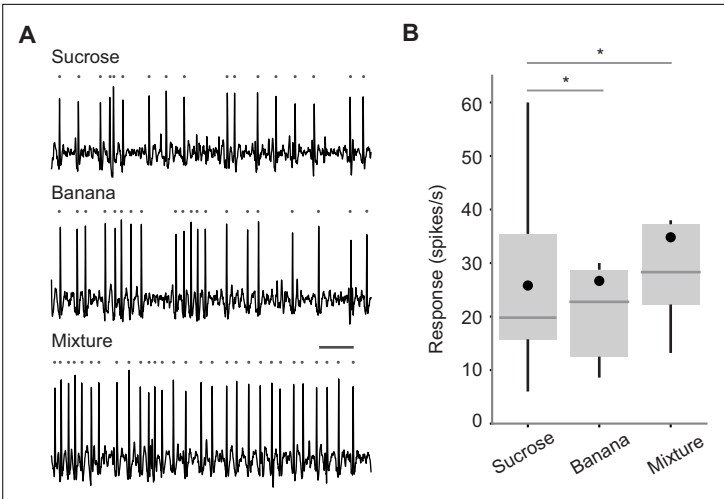

**Figure 5.** Odor-taste integration in single taste sensillum. (**A**) Example responses of an L-type taste sensillum to sucrose (0.25%, top), banana odor (0.1%, middle), and the mixture of the two (bottom). Traces show the activity between 200 and 700 ms after the stimulus onset. Each gray dot indicates a spike. Bar indicates 50 ms. (**B**) Average responses of L-type sensilla. Responses to the mixture are significantly larger than those to components (n=14 flies, p=0.001, repeated measures ANOVA followed by paired t-tests with Bonferroni correction, p=0.0030 and 0.0032 for sucrose and banana compared to mixture). Box plots indicate the median (gray line), mean (black dot), quartiles (box), and 5–95% range (bar). Asterisk indicates p<0.05.

The online version of this article includes the following figure supplement(s) for figure 5:

**Figure supplement 1.** Correlation between odor and taste responses in individual sensilla.

We found that the presence of banana odor increased the amount of sucrose consumption in intact flies (*Figure 6F*). Critically, this effect was observed in olfactory organs-removed flies as well (*Figure 6F*), indicating the contribution of the gustatory system. A similar result was observed with ethyl butyrate, and a slight, though not significant, increase was also observed with 4-methylcyclohexanol (*Figure 6G*). These results suggest that GRNs serve as an initial node for odor-taste multisensory integration that shapes feeding behavior (*Figure 6H*).

## Discussion

In terrestrial animals, it has been considered that there is a clear division of labor between the two types of chemosensory organs in sensing stimuli with distinct physical properties; the olfactory and gustatory organs sense volatile and non-volatile chemicals, respectively. Therefore, although odor-ants and tastants are defined by the identity of cognate organs, they are used interchangeably with volatile and non-volatile chemicals. However, we found using calcium imaging and electrophysiological recording that GRNs directly detect odorants, and this drives a classical taste-induced feeding behavior, PER, as well as increased food consumption. A previous study has also hinted that odors alone evoke PER (*Oh et al., 2021*). Thus, volatile chemicals are within the molecular receptive range of GRNs and should also be regarded as tastants based on the conventional definition. Critically, this indicates that GRNs are engaged in non-contact chemo-sensing as well, a behavioral strategy distinct from contact-based chemo-sensing.

A study applying base recording, an electrophysiological technique to record from individual taste sensilla, had reported that I-type sensilla do not respond to odorant molecules dissolved in solution (*Hiroi et al., 2008*). However, a recent study using the same method found that S-, I-, and L-type sensilla respond to a variety of odors applied as vapor (*Dweck and Carlson, 2023*). Here, we found that GRNs respond to odors applied in both ways in agreement with *Dweck and Carlson, 2023*. Although the reason behind the discrepancy is not entirely clear, one possibility is the difference between the preparation used in different studies; whereas Hiroi et al. used a severed head preparation, as well as Dweck and Carlson used immobilized, intact flies. Because we noticed in preliminary experiments that stimulus-evoked GRN spikes become weaker over time in a hea- only preparation

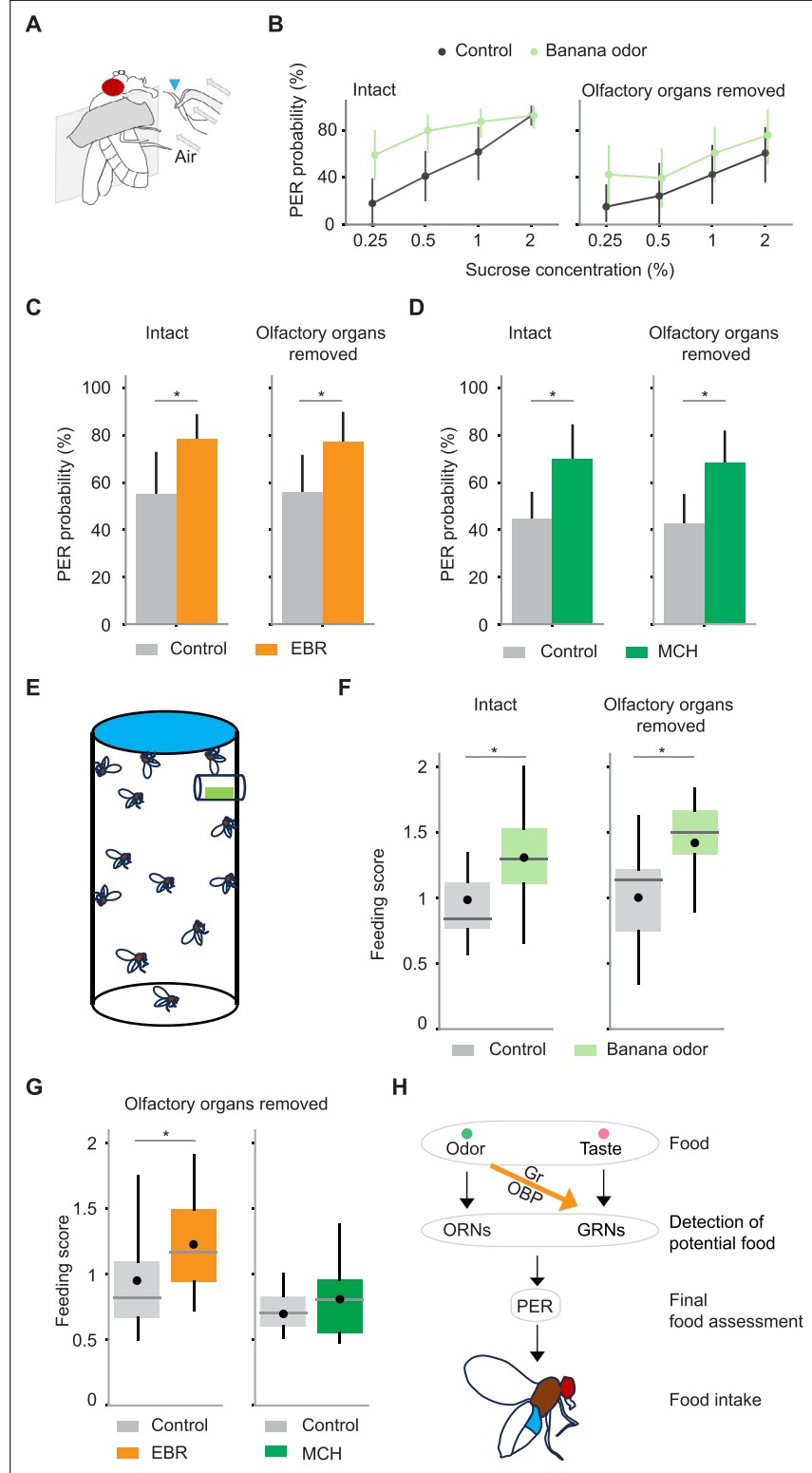

**Figure 6.** Gustatory receptor neurons (GRNs) integrate multisensory input to enhance proboscis extension reflex (PER) and food consumption. (**A**) Schematic for examining PER in response to odor-taste multimodal stimuli. Stimuli are presented to the labellum with a wick (arrowhead) immersed in a sucrose solution with or without odors. A constant air stream was applied from behind the wick. (**B**) PER probability in response to sucrose at different concentrations with (green) or without (black) banana odor in wild-type flies. Odor concentration was $10^{-4}$. Banana odor enhances sucrose-evoked PER not only in intact flies (n=13 and 11 for control and banana odor, p=4.6e-4 and

*Figure 6 continued on next page*

*Figure 6 continued*

2.1e-7 for odor identity and sucrose concentration factors, Scheirer–Ray–Hare test) but also in flies without olfactory organs (n=11 and 11 for control and banana odor, p=0.035 and 0.0070 for odor identity and sucrose concentration factors, Scheirer–Ray–Hare test). Error bar, standard error of the mean. (**C**) PER probability in response to sucrose is increased with ethyl butyrate (EBR) in both intact and olfactory organs-removed flies (n=25 and 25 for intact and olfactory organs-removed flies, p=0.0052 and 0.0068 for intact and olfactory organ-removed flies, Wilcoxon signed-rank test). (**D**) Same as in C, but for 4-methylcyclohexanol (MCH) (n=21 and 36, p=0.0072 and 0.00074 for intact and olfactory organs-removed flies, Wilcoxon signed-rank test). The concentration of odors and sucrose was $10^{-4}$ and 0.25% for **C** and **D**. Error bar, standard error of the mean. (**E**) Schematic for quantifying sucrose consumption in the presence or absence of odors. A filter paper on the top was impregnated with sucrose solution containing blue dye and a small piece of filter paper placed in the mesh tube near the top was impregnated with odor or control solution (see Methods). (**F**) The amount of sucrose consumed by flies with or without the presence of banana odor was quantified with a feeding score (see Methods). The amount of feeding was significantly larger in the presence of odor for both intact (n=20 and 22 for control and banana odors, p=0.005, Wilcoxon signed-rank test) and olfactory organs-removed flies (n=13 and 15 for control and banana odors, p=0.007, Wilcoxon signed-rank test). (**G**) The same as in F but with EBR and MCH odors and only for olfactory organs-removed flies (EBR experiment, n=16 and 17 for control and EBR, p=0.03, Wilcoxon signed-rank test; MCH experiment, n=11 and 14 for control and MCH, p=0.24, Wilcoxon signed-rank test). (**H**) Summary of mechanisms underlying PER and food intake in a multisensory environment. Error bars are the standard error of the mean in bar plots. Box plots indicate the median (gray line), mean (black dot), quartiles (box), and 5–95% range (bar). Asterisk indicates p<0.05.

where hemolymph circulation is lost, the physiological condition of sensilla may be one of the factors underlying the discrepancy. Another possibility is a difference in genotype or genetic background. We found that the strength of odor-evoked PER is different between genetic backgrounds with w[1118] exhibiting weaker PER than Dickinson wild-type (*Figure 1—figure supplement 3*), implying that odor responses of GRNs vary across genetic backgrounds. Furthermore, odor tuning of sugar-sensing GRNs was different between our study and *Dweck and Carlson, 2023*, which used different genetic lines. Although we have observed odor-evoked PER in *Drosophila simulans* as well, further study is required to examine the generality as well as the variability of GRN odor responses across genotypes and species.

Our results showed that odor responses of Gr5a GRNs are mediated by the Gr5a receptor. This may not be totally enigmatic as gustatory and olfactory receptor genes belong to the same ancient superfamily sharing a motif in the transmembrane region (*Robertson et al., 2003*; *Scott et al., 2001*), and some gustatory receptors expressed outside of GRNs function in odor (*Jones et al., 2007*; *Kwon et al., 2007*), temperature (*Ni et al., 2013*), and light detection (*Montell, 2021*; *Xiang et al., 2010*). However, Gr5a is unique in that the same receptor can detect stimuli of two different sensory modalities. Furthermore, we found that OBPs enriched in the labella modulate odor-evoked PER, suggesting that OBPs in this taste organ not only link tastants with gustatory receptors (*Jeong et al., 2013*; *Swarup et al., 2014*), but also mediate detection of odors.

PER was evoked by all the tested odors albeit to different extents, and this likely reflects the broad odor tuning of Gr5a GRNs that drive PER (*Figure 3*). The question, then, is why might it be useful to exhibit PER to such a wide variety of odors including those that are normally aversive? A likely benefit can be discussed by considering foraging behavior under natural settings. During foraging in the air, olfactory receptor neurons that are more sensitive to odors locate a potential food source together with other sensory systems. After landing on the potential food, GRNs on the legs join to assess its edibility. In parallel, GRNs on the labella sense odors and enhance the probability of PER. Given that the olfactory system most likely has already guided the animal to palatable food, keeping the GRNs broadly tuned to odors is an efficient mechanism to enhance PER. Once PER brings the labella into contact with the food, GRNs act as a multisensory integrator to make a decision on food intake (*Figure 6*). Enhancing PER to aversive odors might also be adaptive, as animals often need to carry out the final check by tasting a trace amount of potentially dangerous substances to confirm that those should not be consumed. Because inputs from olfactory and gustatory organs interact centrally as well (*Oh et al., 2021*; *Shiraiwa, 2008*; *Figure 2A*), this indicates that a unified chemosensory experience is built through layered integration in different parts of the body.

Although odor detection by GRNs through gustatory receptors has not previously been reported outside of *Drosophila melanogaster*, olfactory receptors are expressed in the proboscis in multiple

insects (*Haverkamp et al., 2016*; *Kwon et al., 2006*; *Xia and Zwiebel, 2006*). A recent study reported that olfactory receptors are also expressed and functional in cultured human fungiform and mouse taste papilla cells (*Malik et al., 2019*), implying that peripheral odor-taste integration may be a process incorporated in various species. These and our results reveal that GRNs should be further studied not as gustatory neurons but more broadly as chemosensory integrators.

# Materials and methods

Key resources table

| Reagent type (species) or resource | Designation | Source or reference | Identifiers | Additional information |
|---|---|---|---|---|
| Chemical compound, drug | 2-pentanone | Wako | Cat#: 133–03743; CAS: 107-87-9 | |
| Chemical compound, drug | Banana essence | Narizuka corporation | N/A | |
| Chemical compound, drug | Mineral oil | nacalai tesque | Cat#: 23334–85; CAS: 64741-97-5 | |
| Chemical compound, drug | Ethyl butyrate | Sigma-Aldrich | Cat#: E15701; CAS: 105-54-4 | |
| Chemical compound, drug | Isopentyl acetate | Sigma-Aldrich | Cat#: 016–03646; CAS: 123-92-2 | |
| Chemical compound, drug | 4-methylcyclohexanol | Sigma-Aldrich | Cat#: 153095; CAS: 589-91-3 | |
| Chemical compound, drug | 3-octanol | Tokyo Chemical Industry | Cat#: O0121; CAS: 589-98-0 | |
| Chemical compound, drug | Benzaldehyde | Sigma-Aldrich | Cat#: 418099; CAS: 100-52-7 | |
| Chemical compound, drug | 1-hexanol | Tokyo Chemical Industry | H0130 | |
| Chemical compound, drug | Ethyl acetate | Tokyo Chemical Industry | A0030 | |
| Chemical compound, drug | Yeast | nacalai tesque | M1E8921 | |
| Chemical compound, drug | Fenchone | Tokyo Chemical Industry | F0164 | |
| Chemical compound, drug | Isoamyl alcohol | Tokyo Chemical Industry | I0289 | |
| Chemical compound, drug | Phenethyl alcohol | Tokyo Chemical Industry | P0084 | |
| Chemical compound, drug | TCC | Sigma-Aldrich | C0330 | |
| Antibody | Anti-GFP | Nacalai Tesque | 00404–84 | |
| Antibody | Anti-rat CF488A | Biotium | 20023 | |
| Antibody | Anti-mouse CF633 | Biotium | 20120 | |
| Chemical compound, drug | phosphate-buffered saline | Nacalai Tsque | 2757531 | |
| Chemical compound, drug | 4% paraformaldehyde in PBS | Nacalai Tesque | 915414 | |
| Chemical compound, drug | Goat serum | Invitrogen | 50197Z | |
| Chemical compound, drug | Triton X-100 | nacalai tesque | 3550102 | |
| Chemical compound, drug | Vectashield | Vector laboratories | H-1000 | |
| Strain, strain background (*D. melanogaster*) | Dickinson wild-type | Gift from Michael Dickinson | | |
| Strain, strain background (*D. simulans*) | *D. simulans* | Kyoto *Drosophila* Stock Center | DGGR #900001 | |
| Strain, strain background (*D. melanogaster*) | *Tubulin-GAL4* | Bloomington | BDSC #5138 | |

*Continued on next page*

*Continued*

| Reagent type (species) or resource | Designation | Source or reference | Identifiers | Additional information |
|---|---|---|---|---|
| Strain, strain background (*D. melanogaster*) | *Orco-GAL4* | Bloomington | BDSC #26818 | |
| Strain, strain background (*D. melanogaster*) | *Gr66a-GAL4* | Bloomington | BDSC #57670 | |
| Strain, strain background (*D. melanogaster*) | *Ir94e-GAL4* | Bloomington | BDSC #60725 | |
| Strain, strain background (*D. melanogaster*) | *Ppk28-GAL4* | Bloomington | BDSC #60725 | |
| Strain, strain background (*D. melanogaster*) | *UAS-GCaMP6s* | Bloomington | BDSC #42746 | |
| Strain, strain background (*D. melanogaster*) | *UAS-GCaMP6s* | Bloomington | BDSC #42749 | |
| Strain, strain background (*D. melanogaster*) | *UAS-Obp49a RNAi* | Vienna *Drosophila* Resource Center | VDRC #330599 | |
| Strain, strain background (*D. melanogaster*) | *UAS-Obp19b RNAi* | Vienna *Drosophila* Resource Center | VDRC #1823 | |
| Strain, strain background (*D. melanogaster*) | *UAS-Obp56g RNAi* | Vienna *Drosophila* Resource Center | VDRC #23206 | |
| Strain, strain background (*D. melanogaster*) | *UAS-Obp56h RNAi* | Vienna *Drosophila* Resource Center | VDRC #102562 | |
| Strain, strain background (*D. melanogaster*) | *UAS-Obp18a RNAi* | Vienna *Drosophila* Resource Center | VDRC #101628 | |
| Strain, strain background (*D. melanogaster*) | *UAS-Obp83c RNAi* | Vienna *Drosophila* Resource Center | VDRC #106866 | |
| Strain, strain background (*D. melanogaster*) | *UAS-Obp57d/e RNAi* | Vienna *Drosophila* Resource Center | VDRC #101783 | |
| Strain, strain background (*D. melanogaster*) | *UAS-Obp57e RNAi* | Vienna *Drosophila* Resource Center | VDRC #105001 | |
| Strain, strain background (*D. melanogaster*) | *UAS-Obp57a/c RNAi* | Vienna *Drosophila* Resource Center | VDRC #107489 | |
| Strain, strain background (*D. melanogaster*) | $w^{1118}$ | Vienna *Drosophila* Resource Center | VDRC #60000 | |
| Strain, strain background (*D. melanogaster*) | $w^{1118}$;P{VDRCsh60200attP40 | Vienna *Drosophila* Resource Center | VDRC #60200 | |
| Strain, strain background (*D. melanogaster*) | y,$w^{1118}$;P{attP,y[+],w[3']} | Vienna *Drosophila* Resource Center | VDRC #60100 | |
| Strain, strain background (*D. melanogaster*) | *Gr5a-GAL4* | Gift from Kristin Scott | N/A | |
| Strain, strain background (*D. melanogaster*) | *ΔEP(x)–5* | Gift from Anupama Dahanukar | N/A | |
| Strain, strain background (*D. melanogaster*) | *EP(x)496* | Gift from Anupama Dahanukar | N/A | |
| Strain, strain background (*D. melanogaster*) | *UAS-Kir2.1$^{AAE}$-GFP* | Gift from Graeme Davis | N/A | |
| Software, algorithm | Fiji | https://fiji.sc/ | RRID:SCR_002285 | |
| Software, algorithm | MATLAB | MathWorks | RRID:SCR_001622 | |
| Software, algorithm | R | R Project for Statistical Computing | RRID:SCR_001905 | |

*Continued on next page*

*Continued*

| Reagent type (species) or resource | Designation | Source or reference | Identifiers | Additional information |
|---|---|---|---|---|
| Software, algorithm | Python | Python Software | RRID:SCR_008394 | |
| Software, algorithm | ZEISS ZEN Imaging Software | Carl Zeiss, Oberkochen, Germany | RRID:SCR_013672 | |
| Software, algorithm | DeepLabCut v2.0 | *Mathis et al., 2018*; *Mathis, 2021*; https://github.com/DeepLabCut/Docker4DeepLabCut2.0 | | |

## Experimental model and subject details

### *Drosophila* strains

Flies were maintained on standard cornmeal agar under a 12 hr light, 12 hr dark cycle at 25 °C. All experiments were performed on adult females 2–6 days after eclosion. Flies were obtained as described in the key resources table.

Fly stocks used in this study are as follows: Dickinson wild-type (Michael Dickinson), *w[1118]; P{y[+t7.7] w[+mC]=GAL4.1Uw}attP2* (Bloomington #68384), *w[1118]; P{y[+t7.7] w[+mC]=p65.AD.Uw} attP40; P{y[+t7.7] w[+mC]=GAL4.DBD.Uw}attP2* (Bloomington #79603), *D.simulans* (Kyoto DGGR #900001), *tubulin-GAL4* (Bloomington #5138 *Lee and Luo, 1999*), *Orco-GAL4* (Bloomington #26818 *Larsson et al., 2004*), *Gr66a-GAL4* (Bloomington #57670 *Kwon et al., 2011*), *Ir94e-GAL4* (Bloomington #60725 *Jaeger et al., 2018*), *Ppk28-GAL4* (Bloomington #93020 *Cameron et al., 2010*), *UAS-GCaMP6s* (Bloomington #42746, 42749 *Chen et al., 2013*), *UAS-Obp49a RNAi* (VDRC #330599 *Dietzl et al., 2007*), *UAS-Obp19b RNAi* (VDRC #1823), *UAS-Obp56g RNAi* (VDRC #23206), *UAS-Obp56h RNAi* (VDRC #102562), *UAS-Obp18a RNAi* (VDRC #101628), *UAS-Obp83c RNAi* (VDRC #106866), *UAS-Obp57d/e RNAi* (VDRC #101783), *UAS-Obp57e RNAi* (VDRC #105001), *UAS-Obp57a/c RNAi* (VDRC #107489), three OBP RNAi control lines *w[1118]* (VDRC #60000, control for *UAS-Obp19b RNAi* and *UAS-Obp56g RNAi*), *w[1118];P{VDRCsh60200attP40* (VDRC #60200, control for *UAS-Obp49a RNAi*), *y,w[1118];P{attP,y[+],w[3']}* (VDRC #60100, control for other *UAS-Obp RNAi* lines), *Gr5a-GAL4* (Kristin Scott), *Gr5a* mutant (Anupama Dahanukar *Dahanukar et al., 2007*, *ΔEP(x)–5*) and its control (Anupama Dahanukar *Dahanukar et al., 2007*, *EP(x)496*), and *UAS-Kir2.1[AAE]-GFP* (Graeme Davis *Paradis et al., 2001*). All the lines except for *w[1118]*, *D. simulans*, *Gr5a* mutant, its control, and the UAS-RNAi lines were backcrossed for six generations to the Dickinson wild-type (*Dickinson, 1999*).

Detailed genotypes of flies used in each experiment are listed in *Table 1*.

## Examination of odor-evoked PER

Virgin females that had been raised on food for 1–3 days were starved for 24–28 hr in vials with water supplied through a wet piece of Kimwipe. Individual flies were briefly anesthetized on ice and their dorsal side of the thorax was attached to a cover glass with ultraviolet-curing adhesive (NOA 63, Norland), after which the flies were allowed to rest for an hour. Prior to recording odor-evoked PER, flies were water-satiated until they did not consume any more. Subsequently, 100 mM sucrose solution, which acted as a positive control stimulus, was applied either on the fly's legs or proboscis (without letting the fly consume it) to examine if the fly could exhibit tastant-evoked PER. Flies were discarded if they did not exhibit PER to this stimulus. Flies showing excessive spontaneous PER before the assay were also discarded. The tethered fly was positioned horizontally in air facing an odor delivery tube (*Figure 1A*) except for the experiment in Figure S2 where the fly was positioned vertically to mimic the preparation used in a previous study (*Oh et al., 2021*). The tip of the odor tube was placed 10 mm away from the fly. A monochrome camera (Lu070M, Lumenera Corporation) taking a lateral view of the fly at 20 Hz was used to record the movement of proboscis in response to odors.

To examine the contribution of olfactory organs to odor-evoked PER (*Figures 2 and 6*), the third antennal segments and maxillary palps were removed with forceps while the flies were anesthetized on ice. After the surgery, the flies were given an hour to recover before the experiment.

To examine the contribution of GRNs on wings and legs to odor-evoked PER (Figure S5), wings and tarsal segments were removed with forceps while the flies were anesthetized on ice. After the surgery, the flies were given an hour to recover before the experiment.

**Table 1.** Genotypes used to generate data presented in each figure.

| Figure | Genotype |
| --- | --- |
| *Figures 1B–G, 2A and 4–6, Figure 1—figure supplement 1, Figure 1—figure supplement 2, Figure 2—figure supplement 1, Figure 3—figure supplement 3A* | Dickinson Wild-type |
| *Figure 1—figure supplement 3A* | $w^{1118}$; $P\{y[+t7.7]\ w[+mC]=GAL4.1Uw\}attP2$ |
| *Figure 1—figure supplement 3B* | $w^{1118}$; $P\{y[+t7.7]\ w[+mC]=p65.AD.Uw\}attP40$; $P\{y[+t7.7]\ w[+mC]=GAL4.DBD.Uw\}attP2$ |
| *Figure 1—figure supplement 3C* | D. simulans |
| *Figure 2B* | UAS-Kir2.1/+ |
| *Figure 2C* | Gr5a-Gal4/UAS-Kir2.1 Gr5a-Gal4/+ |
| *Figure 2D* | Gr66a-Gal4/UAS-Kir2.1 Gr66a-Gal4/+ |
| *Figure 2E* | Ppk28-Gal4/UAS-Kir2.1 Ppk28-Gal4/+ |
| *Figure 2F* | Ir94e-Gal4/UAS-Kir2.1 Ir94e-Gal4/+ |
| *Figure 2G* | Gr66a-Gal4/UAS-Kir2.1 UAS-Kir2.1/+ |
| *Figure 2—figure supplement 2* | Orco-Gal4 >UAS-Kir2.1 |
| *Figure 2H* | $\Delta EP(x)$–5 $EP(x)496$ |
| *Figure 2I* | $tubulin$-$Gal4/y,w^{1118}$;$P\{attP,y[+],w[3']\}$ tubulin-Gal4/UAS-Obp57d/e RNAi |
| *Figure 2—figure supplement 3* | $tubulin$-$Gal4/y,w^{1118}$;$P\{attP,y[+],w[3']\}$ |
| *Figure 2J* | $tubulin$-$Gal4/w^{1118}$;$P\{VDRCsh60200\}attP40$ tubulin-Gal4/UAS-Obp49a RNAi |
| *Figure 2K* | $tubulin$-$Gal4/w^{1118}$ tubulin-Gal4/UAS-Obp19b RNAi |
| *Figure 2—figure supplement 3* | $tubulin$-$Gal4/w^{1118}$ |
| *Figure 2—figure supplement 3* | tubulin-Gal4/UAS-Obp56g RNAi |
| *Figure 2—figure supplement 3* | tubulin-Gal4/UAS-Obp18a RNAi |
| *Figure 2—figure supplement 3* | tubulin-Gal4/UAS-Obp56h RNAi |
| *Figure 2—figure supplement 3* | tubulin-Gal4/UAS-Obp83c RNAi |
| *Figure 2—figure supplement 3* | tubulin-Gal4/UAS-Obp57e RNAi |
| *Figure 2—figure supplement 3* | tubulin-Gal4/UAS-Obp57a/c RNAi |
| *Figure 3—figure supplement 1J* | UAS-GCaMP6s;Ppk28-Gal4 |
| *Figure 3—figure supplement 1K* | UAS-GCaMP6s;Ir94e-Gal4 |
| *Figure 3A-D, G, Figure 3—figure supplement 1A-C, G, I, Figure 3—figure supplement 2A, Figure 3—figure supplement 3B* | Gr5a-Gal4;UAS-GCaMP6s |
| *Figure 3E, F, Figure 3—figure supplement 1D-F, Figure 3—figure supplement 2B* | UAS-GCaMP6s;Gr66a-Gal4 |
| *Figure 3H* | $\Delta EP(x)$–5;Gr5a-Gal4;UAS-GCaMP6s $EP(x)496$;Gr5a-Gal4;UAS-GCaMP6s |

Individual flies went through an experiment consisting of three blocks. In each block, six odors (see below), two solvent controls (mineral oil and water), and another control stimulus (air) were applied for 2 s per trial in a randomized order with a 15 s inter-trial interval.

## Olfactory stimulation

Odors were delivered with a custom-made, multi-channel olfactometer controlled by a PC as previously described (*Badel et al., 2016*). Briefly, an air stream (300 ml/min) was passed through 4 ml of odor solution diluted with mineral oil (Nacalai Tesque, 23334–85) or water. The concentration of odor was varied between $10^{-4}$, $10^{-2}$, $10^{-1}$, and 0.5 (v/v), but the default was 0.5 unless otherwise noted. The odorized air was further diluted by mixing it with a main air stream (500 ml/min), a small portion of which was delivered frontally to the fly through an outlet placed 10 mm away from the fly. The speed of odorized air flow at the position of the fly was 0.6 m/s. Using the photoionization detector (200B miniPID, Aurora Scientific Inc), the time odors reach the position of the fly was estimated to be 1.1 s after the odor valve opening. The odors used in the study and their abbreviations are as follows: 2-pentanone (2PT, Wako, 13303743), 3-octanol (OCT, Tokyo Chemical Industry, O0121), 4-methylcyclohexanol (MCH, Sigma-Aldrich, 153095), banana essence (Narizuka Corporation), benzaldehyde (BNZ, Sigma-Aldrich, 418099), ethyl butyrate (EBR, Sigma-Aldrich, E15701), and isopentyl acetate (IPA, Wako, 016–03646), 1-hexanol (Tokyo Chemical Industry, H0130), ethyl acetate (Tokyo Chemical Industry, A0030), yeast (nacalai tesque, M1E8921), fenchone (Tokyo Chemical Industry, F0164), isoamyl alcohol (Tokyo Chemical Industry, I0289), phenethyl alcohol (Tokyo Chemical Industry, P0084).

## Examination of multisensory PER

To examine how flies respond to odor-taste multimodal stimuli presented to GRNs in the labella, individual flies were gently attached to a cover glass vertically by wrapping their thorax with a piece of parafilm, and multimodal stimuli were applied by touching the ventral part of the labellum with a wick made of Kimwipe immersed in a sucrose solution with or without odors (*Figure 4A*; *Shiraiwa, 2008*). A constant air stream (0.5–0.7 m/s) was applied with an air pump (eAir6000, GEX) from behind the wick. The legs of the flies were tucked under the parafilm to prevent them from touching the wick. The flies were allowed to rest for an hour after fixation. The concentration of sucrose was varied between 0.25–2% (v/v, *Figure 6B*) or fixed at 0.25% (*Figure 6C and D*) depending on the experiment. When the concentration of sucrose was varied, the stimuli were applied in an ascending order to avoid adaptation. The concentration of the added odor was $10^{-4}$ for all the tested stimuli. Each stimulus was applied three times with an inter-trial interval of 15 s. PER was manually scored when the proboscis was fully extended within ~2 s from the stimulation.

## Quantification of PER

A markerless pose estimation algorithm, DeepLabCut v2.0 was used to quantify the movement of the proboscis. The position and orientation of three segments of the proboscis, the rostrum, the haustellum, and the labellum were characterized by tracking six points, namely the proximal end of the antenna, the distal end of the antenna, the rostrum apex, the rostrum-haustellum joint, the haustellum-labellum joint, and the distal end of the labellum (*Figure 1B and C*). These six points were manually labeled on the lateral view of the fly to generate the training dataset, which consisted of 1465 labeled frames in video data from 19 flies. The ResNet50-based pose estimation neural network was trained for about 200,000 iterations, after which the six points were automatically tracked in all the frames in video data.

Following the pose estimation, data were analyzed using custom code written in Python. To detect proboscis extensions, the rostrum angle (the angle made by the line passing through the proximal and distal ends of the antenna and the line along the rostrum), the haustellum angle (the angle between the rostrum and the haustellum), and the labellum angle (the angle between the haustellum and the labellum) were calculated over time (see *Figure 1B*). The baseline of each proboscis angle was calculated as the $25^{th}$ percentile of a rolling 5 s time window. The proboscis extension was defined as PER if the haustellum angle exceeded 100° because it corresponded to full extension by visual inspection. PER duration was defined as the time during which the fly exhibited PER in each second. Integrated PER duration was defined as the sum of PER duration over 4 s starting 1 s after the odor onset

considering the time that odor requires to reach the fly (see earlier section on Olfactory stimulation) and the time period during which the majority of PER occurred. These values were averaged across three trials for each odor in each fly. PER probability was defined as the percentage of trials in which PER was observed for each stimulus in each fly.

Because repetitive, spontaneous PER occurring at regular intervals represented an abnormal condition, flies showing such repetitive, spontaneous PER in more than 9 trials were excluded from further analysis.

To examine if the movement of the proboscis is similar between odor-evoked and tastant-evoked PER, as well as between PER evoked by different odors, temporal sequences of the rostrum and the haustellum angles during PER and partial proboscis extensions were clustered with K-medoids clustering (*Park and Jun, 2009*). PER and partial proboscis extensions were detected from the baseline subtracted rostrum angle using hysteresis thresholding with a lower threshold of 5° and an upper threshold of 15°. Because the length of a temporal sequence is different between PER, the data were converted to a distance metric using dynamic time warping (*Mearns et al., 2020*; *Sakoe and Chiba, 1978*) with a warping window of 0.25 s prior to clustering. The optimal number of clusters was determined using the elbow method.

## Immunohistochemistry

To examine the expression pattern of Gal4 driver lines, we performed immunohistochemistry as described previously (*Badel et al., 2016*) using rat anti-GFP (1:1,000, Nacalai Tesque, 04404–84) and mouse nc82 (1:20, Developmental Studies Hybridoma Bank at the University of Iowa) as primary antibodies, and anti-rat CF488A (1:250, Biotium, 20023), and anti-mouse CF633 (1:250, Biotium, 20120) as secondary antibodies. Brains were dissected out from the head capsule in phosphate-buffered saline (PBS, nacalai tesque, 2757531), fixed with 4% paraformaldehyde in PBS (nacalai tesque, 915414) for 90 min on ice, and incubated in blocking solution containing 5% normal goat serum (Invitrogen, 50197Z) in PBST (0.2% Triton X-100 (nacalai tesque, 3550102) in PBS) for 30 min. Primary antibodies were then added and incubated at 4°C for ~48 hr. After removing antibodies and washing for over an hour, the brains were incubated in a solution of secondary antibodies at 4°C for 24 hr. The brains were immersed in Vectashield (Vector Laboratories, H-1000), sealed with a cover glass, and imaged with a confocal microscope (FV3000, Olympus). Images were analyzed with Fiji (*Schindelin et al., 2012*).

## Fly preparation for calcium imaging

Individual flies starved for 24–28 hr with water were anesthetized on ice, and their dorsal side of the thorax was fixed to a piece of parafilm with ultraviolet-curing adhesive (NOA 63, Norland). The legs of the fly were covered with another piece of parafilm. The fly was subsequently attached to a custom-made recording plate in a vertical position such that the anterior part of the head was accessible through a small hole on the recording plate (*Figure 3A*). The proboscis was pulled out gently with forceps and immobilized in an extended position using a strip of parafilm with the labella exposed to the air. After covering the head with saline containing 103 mM NaCl, 3 mM KCl, 5 mM N-tris (hydroxymethyl) methyl-2-aminoethane-sulfonic acid, 8 mM trehalose, 10 mM glucose, 26 mM NaHCO$_3$, 1 mM NaH$_2$PO$_4$, 1.5 mM CaCl$_2$, and 4 mM MgCl$_2$ (osmolarity adjusted to 270–275 mOsm), the antennae and the associated cuticle were removed to expose the subesophageal zone. Saline was bubbled with 95% O$_2$/5% CO$_2$ and perfused at a rate of 2 ml/min during the recording. To examine the contribution of ORNs in the maxillary palps and GRNs in the labellum, each of these sensory organs were covered with UV-curable glue (*Figure 3D and F*).

## Two-photon calcium imaging

Calcium imaging of GRN axons in the subesophageal zone was conducted using a two-photon microscope (LSM 7 MP, Zeiss) equipped with a piezo motor (P-725.2CD PIFOC, PI) that drives a water immersion objective lens (W Plan-Apochromat, 20 x, numerical aperture 1.0) along the z-axis. The fluorophore was excited with a titanium:sapphire pulsed laser (Chameleon Vision II, Coherent) mode-locked at 930 nm. The laser power measured at the back aperture of the objective lens was below 20 mW. Fluorescence was collected with a GaAsP detector through a bandpass emission filter (BP470-550). Five optical slices separated by 5 µm were scanned every 500 ms. The odor delivery system was

identical to that used in behavioral experiments. Sucrose was presented by a syringe whose movement was controlled by an actuator.

## Image processing

Calcium imaging data were analyzed using custom code written in MATLAB (MathWorks). Images were registered within and across trials to correct for movement in the x-y plane as well as in depth by determining the shift along three dimensions that maximizes the correlation between the images. The region of interest (ROI) was set to cover the GRN axons in the subesophageal zone. The size of ROI was 80×60 µm for Gr5a GRNs, 60×40 µm for Gr66a GRNs, 80×60 µm for Ir94e GRNs, and 80×40 µm for Ppk28 GRNs, respectively. The average pixel intensity within the ROI was calculated for each time frame. The average of five frames preceding a stimulus was used as the baseline signal to calculate ΔF/F for each time frame. The peak stimulus response was quantified by averaging ΔF/F across five frames at the peak, followed by averaging across three trials for each stimulus. Odor stimulation began at frame 11, and the frames used for peak quantification were 12–16.

## Quantifying the relationship between GRN activity and PER

A linear model in *Figure 3G* was generated using Python scikit-learn to quantify the relationship between GRN odor responses and PER. The weights for Gr5a and Gr66a GRNs were estimated using the LinearRegression function. The performance of the model was quantified by calculating the coefficient of determination.

## Electrophysiological recording from taste sensilla on the proboscis

Odor and taste responses of individual taste sensilla on the proboscis were measured using tip recording (*Hodgson et al., 1955*). Virgin females that had been raised on food for 1–2 days were starved for 24–28 hr in vials with water supplied through a wet piece of Kimwipe. Individual flies were immobilized at the end of a pipette tip. The proboscis was fixed at an extended position using ultraviolet-curing adhesive (NOA 63, Norland). A saline-filled reference electrode was inserted into the eye. Individual taste sensilla were stimulated for ~2 s with a capillary electrode containing the stimulus and an electrolyte, 30 mM TCC. The recording electrode was connected to an amplifier (Multiclamp 700B, Molecular Devices). Signals were bandpass filtered at 100–1000 Hz, and digitally sampled at 10 kHz using a data acquisition module (NI cDAQ9178, NI 9215, National Instruments). The intensity of the response was measured by counting the number of spikes generated between 200 and 700ms after the stimulus contact following the convention to avoid the contamination of motion artifact (*Dahanukar and Benton, 2023*; *Delventhal et al., 2014*; *Hiroi et al., 2002*). Responses to sucrose, banana odor, MCH, and TCC were recorded from L- and I-type sensilla. Sucrose and odors were dissolved in TCC (30 mM) solution. Each stimulus was applied to the same sensillum for two or three times, with 1 min inter-stimulus interval. To examine odor-taste integration in each sensillum, banana odor and sucrose solution were presented individually or simultaneously in a mixture. Recordings were made from L2 and L6 sensilla. Each stimulus was presented twice, with 1 min inter-stimulus interval, in a random order.

## Food consumption assay

Food consumption was quantified in principle as described in a previous study (*Reisenman and Scott, 2019*). Briefly, virgin females that had been raised on food for 1–2 days were starved for 24–28 hr in vials with water supplied through a wet piece of Kimwipe. Flies were then transferred to a custom-made vial (n=10–15 per vial) with a filter paper (3 cm in diameter) on the top and a small mesh tube attached laterally near the top (*Figure 6E*). The filter paper on the top was impregnated with sucrose solution (0.1%, 150 µl) containing blue dye blue, Kyoritsu-Foods, and a small piece of filter paper (0.5×1 cm) placed in the mesh tube was impregnated with odor (1%, 10 µl) or control solution. Flies were allowed to consume sucrose solution for 5 min after which the vials were placed in the freezer (–20°C) for at least 1 hr. The extent of sucrose consumption by individual flies was quantified under the microscope based on the amount of blue dye in the abdomen, which was scored using the five-point scale ranging from 0 to 2 (*Reisenman and Scott, 2019*). The scorer was blind to the experimental condition (whether the filter paper in the mesh tube was impregnated with odor or control solvent).

**Table 2.** Summary of statistical tests.

| Results of multi-factorial statistical tests | | P-values for individual factors and their interaction | | |
|---|---|---|---|---|
| **Figure** | **Statistical test** | **Concentration** | **Identity** | **Interaction** |
| *Figure 1E* | Scheirer–Ray–Hare | 3.0E-05 | 2.0E-07 | 2.0E-10 |
| *Figure 1G* | Scheirer–Ray–Hare | 3.6E-13 | 3.5E-11 | 2.8E-21 |
| | | **Method** | **Odor** | **Interaction** |
| *Figure 1—figure supplement 1C* | Scheirer–Ray–Hare | 0.94 | 5.5E-03 | 2.8E-03 |
| | | **State** | **Odor** | **Interaction** |
| *Figure 2—figure supplement 1A* | Scheirer–Ray–Hare | 0.88 | 5.6E-03 | 1.8E-03 |
| *Figure 2—figure supplement 1B* | Scheirer–Ray–Hare | 1.0E-06 | 0.74 | 2.7E-03 |
| | | **Genotype** | **Odor** | **Interaction** |
| *Figure 2A* | Scheirer–Ray–Hare | 0.0079 | 0.10 | 0.0020 |
| *Figure 2B and C* (comparison to UAS control) | Scheirer–Ray–Hare | 4.9E-05 | 0.12 | 3.5E-05 |
| *Figure 2B and D* (comparison to UAS control) | Scheirer–Ray–Hare | 0.49 | 0.0014 | 6.6E-04 |
| *Figure 2B and E* (comparison to UAS control) | Scheirer–Ray–Hare | 0.17 | 0.55 | 0.027 |
| *Figure 2B and F* (comparison to UAS control) | Scheirer–Ray–Hare | 0.11 | 1.2E-05 | 2.0E-07 |
| *Figure 2C* | Scheirer–Ray–Hare | 1.0E-15 | 0.71 | 1.0E-16 |
| *Figure 2D* | Scheirer–Ray–Hare | 0.28 | 0.0012 | 3.0E-05 |
| *Figure 2E* | Scheirer–Ray–Hare | 6.2E-04 | 0.56 | 0.0036 |
| *Figure 2F* | Scheirer–Ray–Hare | 0.14 | 5.6E-09 | 1.7E-08 |
| *Figure 2G* | Scheirer–Ray–Hare | 1.8E-04 | 0.041 | 2.6E-05 |
| *Figure 2H* | Scheirer–Ray–Hare | 1.1E-04 | 0.32 | 0.00015 |
| *Figure 2I* | Scheirer–Ray–Hare | 0.039 | 0.48 | 0.0083 |
| *Figure 2J* | Scheirer–Ray–Hare | 0.0066 | 0.16 | 0.0051 |
| *Figure 2K* | Scheirer–Ray–Hare | 0.24 | 0.26 | 0.0032 |
| *Figure 2—figure supplement 3A and B* | Scheirer–Ray–Hare | 0.58 | 0.0017 | 0.0011 |
| *Figure 2—figure supplement 3C and D* | Scheirer–Ray–Hare | 0.050 | 0.10 | 0.0060 |
| *Figure 2—figure supplement 3C and E* | Scheirer–Ray–Hare | 0.36 | 0.48 | 0.041 |
| *Figure 2—figure supplement 3C and F* | Scheirer–Ray–Hare | 0.18 | 0.063 | 0.011 |
| *Figure 2—figure supplement 3C and G* | Scheirer–Ray–Hare | 1.9E-04 | 0.045 | 8.0E-06 |
| *Figure 2—figure supplement 3C and H* | Scheirer–Ray–Hare | 0.92 | 0.022 | 0.0072 |
| | | **Organ condition** | **Odor** | **Interaction** |
| *Figure 3D* (control vs labella covered) | Mixed two-way ANOVA | 2.6E-15 | 5.1E-03 | 9.8E-03 |
| *Figure 3D* (control vs palps covered) | Mixed two-way ANOVA | 0.69 | 0.055 | 0.91 |
| *Figure 3F* (control vs labella covered) | Mixed two-way ANOVA | 3.0E-16 | 1.1E-07 | 1.9E-07 |
| *Figure 3F* (control vs palps covered) | Mixed two-way ANOVA | 0.48 | 4.2E-11 | 0.90 |
| *Figure 3H* | Mixed two-way ANOVA | 8.0E-06 | 0.037 | 0.070 |
| | | **Odor identity** | **Sucrose concentration** | **Interaction** |
| *Figure 6B* (intact) | Scheirer–Ray–Hare | 4.6E-04 | 2.1E-07 | 0.057 |
| *Figure 6B* (olfactory organs removed) | Scheirer–Ray–Hare | 0.035 | 0.0070 | 0.93 |

For the examination of olfactory organs-removed flies, their third antennal segments and maxillary palps were removed with forceps before starvation.

## Statistical analysis

Statistical analyses were performed using Python (3.8.8) or R (4.0.2). The statistical tests used, and the sample sizes are listed in figure legends and *Table 2*. We predetermined sample sizes not using any statistical methods but based on effect sizes and sample-by-sample variability observed in pilot experiments.

## Acknowledgements

We thank Anupama Dahanukar, Graeme Davis, Michael Dickinson, Craig Montell, Kristin Scott, the Bloomington Stock Center, and the Vienna *Drosophila* Resource Center for flies; RIKEN CBS-Evident Collaboration Center for imaging equipment and software; Bor-Wei Cherng for help with coding; members of the Kazama laboratory for their support and comments on the manuscript. HPW was supported by Grant-in-Aid for JSPS Research Fellow (JP15F15384) and RIKEN Special Postdoctoral Fellowship. This work was supported by a grant from RIKEN, Kao Corporation, Asahi Corporation, Toray (23–6402), Japan Society for the Promotion of Science (KAKENHI Grant JP18H02532, JP18K19502, JP21H04789), and Japan Science and Technology Agency (CREST grant number JP24028095) to HK.

## Additional information

### Funding

| Funder | Grant reference number | Author |
|---|---|---|
| Japan Society for the Promotion of Science | JP15F15384 | Hongping Wei |
| RIKEN | SPDR | Hongping Wei |
| RIKEN | | Hokto Kazama |
| Japan Society for the Promotion of Science | JP18H02532 | Hokto Kazama |
| Japan Science and Technology Agency | JP24028095 | Hokto Kazama |
| Toray | 23-6402 | Hokto Kazama |
| Kao Corporation | | Hokto Kazama |
| Asahi Group Holdings | | Hokto Kazama |
| Japan Society for the Promotion of Science | JP18K19502 | Hokto Kazama |
| Japan Society for the Promotion of Science | JP21H04789 | Hokto Kazama |

The funders had no role in study design, data collection and interpretation, or the decision to submit the work for publication.

### Author contributions

Hongping Wei, Conceptualization, Formal analysis, Supervision, Funding acquisition, Investigation, Visualization, Methodology, Writing – original draft, Writing – review and editing; Thomas Ka Chung Lam, Formal analysis, Investigation, Methodology, Writing – review and editing; Hokto Kazama, Conceptualization, Formal analysis, Supervision, Funding acquisition, Investigation, Writing – original draft, Project administration, Writing – review and editing

### Author ORCIDs

Hongping Wei https://orcid.org/0000-0002-2910-8344
Thomas Ka Chung Lam https://orcid.org/0000-0002-4547-3245

Hokto Kazama  https://orcid.org/0000-0002-3239-1873

Reviewer #1 (Public review): https://doi.org/10.7554/eLife.101440.3.sa1
Reviewer #3 (Public review): https://doi.org/10.7554/eLife.101440.3.sa2
Author response https://doi.org/10.7554/eLife.101440.3.sa3

## Additional files

**Supplementary files**
MDAR checklist

**Data availability**

All data and code used in this paper are deposited in the RIKEN CBS Data Sharing Platform (https://doi.org/10.60178/cbs.20250801-001). All original code has been deposited at GitHub (copy archived at *Wei, 2025*). Any additional information required to analyze the data reported in this study is available from the Lead Contact upon request.

The following dataset was generated:

| Author(s) | Year | Dataset title | Dataset URL | Database and Identifier |
| --- | --- | --- | --- | --- |
| Kazama H | 2025 | Odors drive feeding through gustatory receptor neurons in Drosophila | https://doi.org/10.60178/cbs.20250801-001 | RIKEN, 10.60178/cbs.20250801-001 |

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
