## [Editor Report · eLife Assessment]

This study **convincingly** demonstrates that odors evoke a feeding response in *Drosophila*, mediated by gustatory receptors and observed as a proboscis extension. The evidence is comprehensive, encompassing behavior, functional imaging and electrophysiology. This **important** results on the molecular and cellular basis of multimodal integration across olfaction and gustation will be of interest for the study of chemosensation, sensory biology, and animal behavior.

---

## [Referee Report · Reviewer #1 (Public review)]

Summary:

Odor- and taste-sensing are mediated by two different systems, the olfactory and gustatory systems, and have different behavioral roles. In this study, Wei et al. challenge this dichotomy by showing that odors can activate gustatory receptor neurons (GRNs) in *Drosophila* to promote feeding responses, including the proboscis extension response (PER) that was previously thought to be driven only by taste. While previous studies suggested that odors can promote PER to appetitive tastants, Wei et al. go further to show that odors alone cause PER, this effect is mediated through sweet-sensing GRNs, and sugar receptors are required. The study also shows that odor detection by bitter-sensing GRNs suppresses PER. The authors' conclusions are supported by behavioral assays, calcium imaging, electrophysiological recordings, and genetic manipulations. The observation that both attractive and aversive odors promote PER leaves an open question as to why this effect is adaptive. Overall, the study sheds new light on chemosensation and multimodal integration by showing that odor and taste detection converge at the level of sensory neurons, a finding that is interesting and surprising while also being supported by another recent study (Dweck & Carlson, Sci Advances 2023).

Strengths:

(1) The main finding that odors alone can promote PER by activating sweet-sensing GRNs is interesting and novel.

(2) The study uses video tracking of the proboscis to quantify PER rather than manual scoring, which is typically used in the field. The tracking method is less subjective and provides a higher-resolution readout of the behavior.

(3) The study uses calcium imaging and electrophysiology to show that odors activate GRNs. These represent complementary techniques that measure activity at different parts of the GRN (axons versus dendrites, respectively) and strengthen the evidence for this conclusion.

(4) Genetic manipulations show that odor-evoked PER is primarily driven by sugar GRNs and sugar receptors rather than olfactory neurons. This is a major finding that distinguishes this work from previous studies of odor effects on PER and feeding (e.g., Reisenman & Scott, 2019; Shiraiwa, 2008) that assumed or demonstrated that odors were acting through olfactory neurons.

Weaknesses/Limitations:

(1) Many of the odor effects on behavior or neuronal responses were only observed at very high concentrations. Most effects seemed to require concentrations of at least 10^-2 (0.01 v/v), which is at the high end of the concentration range used in olfactory studies (e.g., Hallem et al., 2004), and most experiments in the paper used a far higher concentration of 0.5 v/v. It is unclear whether these are concentrations that would be naturally encountered by flies. In addition, it is difficult to compare the concentrations used for electrophysiology and behavior given that they are presented in solution versus volatile form.

(2) The timecourse of GRN activation by odors seems quite prolonged (and possibly delayed, depending on the exact timing of odor onset to the fly), and this timecourse is not directly compared with activation by tastes to determine whether it is a property of the calcium sensor or a real difference.

(3) While the overall effect of different conditions is tested using appropriate statistical methods, post-hoc tests are not always used to determine which specific groups are different from each other (e.g., which odors and concentrations elicit significant PER compared to air or mineral oil controls in Fig. 1; which odors show impaired responses without olfactory organs in Fig. 2A).

Discrepancies with previous studies:

These discrepancies are important to note but should not necessarily be considered "weaknesses" of the present study.

(1) It is not entirely clear why PER to odors alone has not been previously reported, especially as this study shows that it is a broad effect evoked by many different odors. Previous studies (Oh et al., 2021; Reisenman & Scott, 2019; Shiraiwa, 2008) tested the effect of odors on PER and only observed enhancement of PER to sugar rather than odor-evoked PER; some of these studies explicitly show no effect of odor alone or odor with low sugar concentration. In the Response to Reviewers, the authors propose that genetic background may explain discrepancies, but this is not discussed much in the paper itself. Differences in behavioral quantification (automated vs. manual scoring, quantification of PER duration versus probability) may also contribute.

(2) The calcium imaging data showing that sugar GRNs respond to a broad set of odors contrasts with results from Dweck & Carlson (Sci Adv, 2023) who recorded sugar neurons with electrophysiology and observed responses to organic acids, but not other odors. This discrepancy is mentioned in the Discussion but the underlying reason is not clear.

---

## [Referee Report · Reviewer #3 (Public review)]

Summary:

Using flies, Kazama et al. combined behavioral analysis, electrophysiological recordings, and calcium imaging experiments to elucidate how odors activate gustatory receptor neurons (GRNs) and elicit a proboscis extension response, which is interpreted as a feeding response.

The authors used DeepLabCut v2.0 to estimate the extension of the proboscis, which represents an unbiased and more precise method for describing this behavior compared to manual scoring.

They demonstrated that the probability of eliciting a proboscis extension increases with higher odor concentrations. The most robust response occurs at a 0.5 v/v concentration, which, despite being diluted in the air stream, remains a relatively high concentration. Although the probability of response is not particularly high it is higher than control stimuli. Notably, flies respond with a proboscis extension to both odors that are considered positive and those regarded as negative.

The authors used various transgenic lines to show that the response is mediated by GRNs. Specifically, inhibiting Gr5a reduces the response, while inhibiting Gr66a increases it in fed flies. Additionally, they find that odors induce a strong positive response in both types of GRNs, which is abolished when the labella of the proboscis are covered. This response was also confirmed through electrophysiological tip recordings.

Finally, the authors demonstrated that the response increases when two stimuli of different modalities, such as sucrose and odors, are presented together, suggesting clear multimodal integration

Strengths:

The integration of various techniques, which collectively supports the robustness of the results.

The assessment of electrophysiological recordings in intact animals, preserving natural physiological conditions.

Weaknesses:

Only highly concentrated odours are capable of evoking positive responses and, even then, the proportion remains relatively low.

The authors have incorporated my suggestions.

---

## [Author Response]

The following is the authors’ response to the original reviews

**Public Review:**

**Reviewer #1 (Public review):**
Summary:Odor- and taste-sensing are mediated by two different systems, the olfactory and gustatory systems, and have different behavioral roles. In this study, Wei et al. challenge this dichotomy by showing that odors can activate gustatory receptor neurons (GRNs) in *Drosophila* to promote feeding responses, including the proboscis extension response (PER) that was previously thought to be driven only by taste. While previous studies suggested that odors can promote PER to appetitive tastants, Wei et al. go further to show that odors alone cause PER, this effect is mediated through sweet-sensing GRNs, and sugar receptors are required. The study also shows that odor detection by bitter-sensing GRNs suppresses PER. The authors' conclusions are supported by behavioral assays, calcium imaging, electrophysiological recordings, and genetic manipulations. The observation that both attractive and aversive odors promote PER leaves an open question as to why this effect is adaptive. Overall, the study sheds new light on chemosensation and multimodal integration by showing that odor and taste detection converge at the level of sensory neurons, a finding that is interesting and surprising while also being supported by another recent study (Dweck & Carlson, Sci Advances 2023).Strengths:(1) The main finding that odors alone can promote PER by activating sweet-sensing GRNs is interesting and novel.(2) The study uses video tracking of the proboscis to quantify PER rather than manual scoring, which is typically used in the field. The tracking method is less subjective and provides a higherresolution readout of the behavior.(3) The study uses calcium imaging and electrophysiology to show that odors activate GRNs. These represent complementary techniques that measure activity at different parts of the GRN (axons versus dendrites, respectively) and strengthen the evidence for this conclusion.(4) Genetic manipulations show that odor-evoked PER is primarily driven by sugar GRNs and sugar receptors rather than olfactory neurons. This is a major finding that distinguishes this work from previous studies of odor effects on PER and feeding (e.g., Reisenman & Scott, 2019; Shiraiwa, 2008) that assumed or demonstrated that odors were acting through olfactory neurons.

We appreciate the reviewer’s positive assessment of the novelty and significance of our work.

Weaknesses/Limitations:(1) The authors may want to discuss why PER to odors alone has not been previously reported, especially as they argue that this is a broad effect evoked by many different odors. Previous studies testing the effect of odors on PER only observed odor enhancement of PER to sugar (Oh et al., 2021; Reisenman & Scott, 2019; Shiraiwa, 2008) and some of these studies explicitly show no effect of odor alone or odor with low sugar concentration; regardless, the authors likely would have noticed if PER to odor alone had occurred. Readers of this paper may also be aware of unpublished studies failing to observe an effect of PER on odor alone (including studies performed by this reviewer and unrelated work by other colleagues in the field), which of course the authors are not expected to directly address but may further motivate the authors to provide possible explanations.

We appreciate the reviewer’s comment. We believe that the difference in genotype is likely the largest reason behind this point. This is because the strength varied widely across genotypes and was quite weak in some strains including commonly used w[1118] empty Gal4 and w[1118] empty spit Gal4 as shown in Figure1- figure supplement 3 (Figure S3 in original submission). However, given that we observed odor-evoked PER in various genotypes (many in main Figures and three in Figure1- figure supplement 3 including *Drosophila simulans*), the data illustrate that it is a general phenomenon in *Drosophila*. Indeed, although Oh et al. (2021) did not emphasize it in the text, their Fig. 1E showed that yeast odor evoked PER at a probability of 20%, which is much higher than the rate of spontaneous PER in many genotypes. Therefore, this literature may represent another support for the presence of odor-evoked PER. We have expanded our text in the Discussion to describe these issues.

Another possibility is our use of DeepLabcut to quantitatively track the kinematics of proboscis movement, which may have facilitated the detection of PER.

(2) Many of the odor effects on behavior or neuronal responses were only observed at very high concentrations. Most effects seemed to require concentrations of at least 10-2 (0.01 v/v), which is at the high end of the concentration range used in olfactory studies (e.g., Hallem et al., 2004), and most experiments in the paper used a far higher concentration of 0.5 v/v. It is unclear whether these are concentrations that would be naturally encountered by flies.

We acknowledge that the concentrations used are on the higher side, suggesting that GRNs may need to be stimulated with relatively concentrated odors to induce PER. Although it is difficult to determine the naturalistic range of odor concentration, it is at least widely reported that olfactory neurons including olfactory receptor neurons and projection neurons do not saturate, and exhibit odor identity-dependent responses at the concentration of 10^-2^ where odor-evoked PER can be observed. Furthermore, we have shown in Figure 6 that low concentration (10^-4^) of banana odor, ethyl butyrate, and 4-methycyclohexanol all significantly increased the rate of odor-taste multisensory PER even in olfactory organs-removed flies, suggesting that low concentration odors can influence feeding behavior via GRNs in a natural context where odors and tastants coexist at food sites. Finally, we note that odors were further diluted by a factor of 0.375 by mixing the odor stream with the main air stream before being applied to the flies as described in Methods.

(3) The calcium imaging data showing that sugar GRNs respond to a broad set of odors contrasts with results from Dweck & Carlson (Sci Adv, 2023) who recorded sugar neurons with electrophysiology and observed responses to organic acids, but not other odors. This discrepancy is not discussed.

As the reviewer points out, Dweck and Carlson (Sci Adv, 2023) reported using single sensillum electrophysiology (base recording) that sugar GRNs only respond to organic acids whereas we found using calcium imaging from a group of axons and single sensillum electrophysiology (tip recording) that these GRNs respond to a wide variety of odors. Given that we observed odor responses using two methods, the discrepancy is likely due to the differences in genotype examined. We now have discussed this point in the text.

(4) Related to point #1, it would be useful to see a quantification of the percent of flies or trials showing PER for the key experiments in the paper, as this is the standard metric used in most studies and would help readers compare PER in this study to other studies. This is especially important for cases where the authors are claiming that odor-evoked PER is modulated in the same way as previously shown for sugar (e.g., the effect of starvation in Figure S4).

For starved flies, we would like to remind the reviewer that the percentage of trials showing PER is reported in Fig. 1E, which shows a similar trend as the integrated PER duration. For fed flies, we have analyzed the percentage of PER and added the result to Figure 2-figure supplement 1C (Figure S4 in original submission).

(5) Given the novelty of the finding that odors activate sugar GRNs, it would be useful to show more examples of GCaMP traces (or overlaid traces for all flies/trials) in Figure 3. Only one example trace is shown, and the boxplots do not give us a sense of the reliability or time course of the response. A related issue is that the GRNs appear to be persistently activated long after the odor is removed, which does not occur with tastes. Why should that occur? Does the time course of GRN activation align with the time course of PER, and do different odors show differences in the latency of GRN activation that correspond with differences in the latency of PER (Figure S1A)?

Following the reviewer’s suggestion, we now report GCaMP responses for all the trials in all the flies (both Gr5a>GCaMP and Gr66a>GCaMP flies), where the time course and trial-to-trial/animal-toanimal variability of calcium responses can be observed (Figure 3-figure supplement 2).

Regarding the second point, we recorded responses to both sucrose and odors in some flies and found that calcium responses of GRNs are long-lasting not only to odors but also to sucrose, as shown in Author response image 1. This may be due in part to the properties of GCaMP6s and slower decay of intracellular calcium concentration as compared to spikes.

**Author response image 1. sa3fig1:** Example calcium responses to sucrose and odor (MCH) in the same fly (normalized by the respective peak responses to better illustrate the time course of responses). Sucrose (blue) and odor (orange) concentrations are 100 mM, and 10^-1^respectively. Odor stimulation begins at 5 s and lasts for 2 s. Sucrose was also applied at the same timing for the same duration although there was a limitation in controlling the precise timing and duration of tastant application. Because of this limitation, we did not quantify the off time constant of two responses.

To address whether the time course of GRN activation aligns with the time course of PER, and whether different odors evoke different latencies of GRN activation that correspond to latencies of PER, we plotted the time course of GRN responses and PER, and further compared the response latencies across odors and across two types of responses in Gr5a>GCaMP6s flies. As shown in Author response image 2, no significant differences were found in response latency between the six odors for PER and odor responses. Furthermore, Pearson correlation between GRN response latencies and PER latencies was not significant (r = 0.09, p = 0.872).

**Author response image 2. sa3fig2:** (A) PER duration in each second in Gr5a-Gal4>UAS-GCaMP6s flies. The black lines indicate the mean and the shaded areas indicate the standard error of the mean. n = 25 flies. (B) Time course of calcium responses (ΔF/F) to nine odors in Gr5a GRNs. n = 5 flies. (C) Latency to the first odor-evoked PER in Gr5a-Gal4>UAS-GCaMP6s flies. Green bar indicates the odor application period. p = 0.67, one-way ANOVA. Box plots indicate the median (orange line), mean (black dot), quartiles (box), and 5-95% range (bar). Dots are outliers. (D) Latency of calcium responses (10% of rise to peak time) in Gr5a GRNs. Green bar indicates the odor application period. p = 0.32, one-way ANOVA. Box plots indicate the median (orange line), mean (black dot), quartiles (box), and 5-95% range (bar). Dots are outliers.

(6) Several controls are missing, and in some cases, experimental and control groups are not directly compared. In general, Gal4/UAS experiments should include comparisons to both the Gal4/+ and UAS/+ controls, at least in cases where control responses vary substantially, which appears to be the case for this study. These controls are often missing, e.g. the Gal4/+ controls are not shown in Figure 2C-G and the UAS/+ controls are not shown in Figure 2J-L (also, the legend for the latter panels should be revised to clarify what the "control" flies are). For the experiments in Figure S5, the data are not directly compared to any control group. For several other experiments, the control and experimental groups are plotted in separate graphs (e.g., Figure 2C-G), and they would be easier to visually compare if they were together. In addition, for each experiment, the authors should denote which comparisons are statistically significant rather than just reporting an overall p-value in the legend (e.g., Figure 2H-L).

We thank the reviewer for the input. We have conducted additional experiments for four Gal4/+controls in Figure 2 and added detailed information about control flies in the figure legend (Figure 2C-F).

For the RNAi flies shown in Figure 2 and Figure 2-figure supplement 3, we used the recommended controls suggested by the VDRC. These control flies were crossed with tubulin-Gal4 lines to include both Gal4 and UAS control backgrounds.

Regarding Figure S5 in original submission (current Figure 2-figure supplement 2), we now present the results of statistical tests which revealed that PER to certain odors is statistically significantly stronger than that to the solvent control (mineral oil) for both wing-removed and wing-leg-removed flies.

For Figure 2C-F, we now plot the results for experimental and control groups side by side in each figure.

Regarding the results of statistical tests, we have provided more information in the legend and also prepared a summary table (supplemental table).

(7) Additional controls would be useful in supporting the conclusions. For the Kir experiments, how do we know that Kir is effective, especially in cases where odor-evoked PER was not impaired (e.g., Orco/Kir)? The authors could perform controls testing odor aversion, for example. For the Gr5a mutant, few details are provided on the nature of the control line used and whether it is in the same genetic background as the mutant. Regardless, it would be important to verify that the Gr5a mutant retains a normal sense of smell and shows normal levels of PER to stimuli other than sugar, ruling out more general deficits. Finally, as the method of using DeepLabCut tracking to quantify PER was newly developed, it is important to show the accuracy and specificity of detecting PER events compared to manual scoring.

A previous study (Sato, 2023, Front Mol Neurosci) showed that the avoidance to 100 μM 2methylthiazoline was abolished, and the avoidance to 1 mM 2MT was partially impaired in Orco>Kir2.1 flies. However, because Orco-Gal4 does not label all the ORNs and we have more concrete results on flies in which all the olfactory organs are removed as well as specific GRNs and Gr are manipulated, we decided to remove the data for Orco>kir2.1 flies and have updated the text and Figure 2 accordingly.

For the Gr5a mutant and its control, we have added detailed information about the genotype in the figure legend and in the Methods. We have used the exact same lines as reported in Dahanukar et al. (2007) by obtaining the lines from Dr. Dahanukar. Dahanukar et al. has already carefully examined that Gr5a mutant loses responses only to certain types of sugars (e.g. it even retains normal responses to some other sugars), demonstrating that Gr5a mutants do not exhibit general deficits.

As for the PER scoring method, we manually scored PER duration and compared the results with those obtained using DeepLabCut in wild type flies for the representative data. The two results were similar (no statistical difference). We have reported the result in Figure1-figure supplement 1C.

(8) The authors' explanation of why both attractive and aversive odors promote PER (lines 249-259) did not seem convincing. The explanation discusses the different roles of smell and taste but does not address the core question of why it would be adaptive for an aversive odor, which flies naturally avoid, to promote feeding behavior.

We have extended our explanation in the Discussion by adding the following possibility: “Enhancing PER to aversive odors might also be adaptive as animals often need to carry out the final check by tasting a trace amount of potentially dangerous substances to confirm that those should not be further consumed.”

**Reviewer #2 (Public review):**
Summary:A gustatory receptor and neuron enhances an olfactory behavioral response, proboscis extension. This manuscript clearly establishes a novel mechanism by which a gustatory receptor and neuron evokes an olfactory-driven behavioral response. The study expands recent observations by Dweck and Carlson (2023) that suggest new and remarkable properties among GRNs in *Drosophila*. Here, the authors articulate a clear instance of a novel neural and behavioral mechanism for gustatory receptors in an olfactory response.Strengths:The systematic and logical use of genetic manipulation, imaging and physiology, and behavioral analysis makes a clear case that gustatory neurons are bona fide olfactory neurons with respect to proboscis extension behavior.Weaknesses:No weaknesses were identified by this reviewer.

We appreciate the reviewer’s recognition of the novelty and significance of our work.

**Reviewer #3 (Public review):**
Summary:Using flies, Kazama et al. combined behavioral analysis, electrophysiological recordings, and calcium imaging experiments to elucidate how odors activate gustatory receptor neurons (GRNs) and elicit a proboscis extension response, which is interpreted as a feeding response.The authors used DeepLabCut v2.0 to estimate the extension of the proboscis, which represents an unbiased and more precise method for describing this behavior compared to manual scoring.They demonstrated that the probability of eliciting a proboscis extension increases with higher odor concentrations. The most robust response occurs at a 0.5 v/v concentration, which, despite being diluted in the air stream, remains a relatively high concentration. Although the probability of response is not particularly high it is higher than control stimuli. Notably, flies respond with a proboscis extension to both odors that are considered positive and those regarded as negative.The authors used various transgenic lines to show that the response is mediated by GRNs.Specifically, inhibiting Gr5a reduces the response, while inhibiting Gr66a increases it in fed flies. Additionally, they find that odors induce a strong positive response in both types of GRNs, which is abolished when the labella of the proboscis are covered. This response was also confirmed through electrophysiological tip recordings.Finally, the authors demonstrated that the response increases when two stimuli of different modalities, such as sucrose and odors, are presented together, suggesting clear multimodal integration.Strengths:The integration of various techniques, that collectively support the robustness of the results.The assessment of electrophysiological recordings in intact animals, preserving natural physiological conditions.

We appreciate the reviewer’s recognition of the novelty and significance of our work.

Weaknesses:The behavioral response is observed in only a small proportion of animals.

We acknowledge that the probability of odor-evoked PER is lower compared to sucrose-evoked PER, which is close to 100 % depending on the concentration. To further quantify which proportion of animals exhibit odor-evoked PER, we now report this number besides the probability of PER for each odor shown in Fig. 1E. We found that, in wild type Dickinson flies, 73% and 68 % of flies exhibited PER to at least one odor presented at the concentration of 0.5 and 0.1.

**Recommendations for the authors:**

**Reviewer #1 (Recommendations for the authors):**
Minor comments/suggestions:- Define "MO" in Figure 1D.

We have defined it as mineral oil in the figure legend.

- Clarify how peak response was calculated for GCaMP traces (is it just the single highest frame per trial?).

We extended the description in the Methods as follows: “The peak stimulus response was quantified by averaging ΔF/F across five frames at the peak, followed by averaging across three trials for each stimulus. Odor stimulation began at frame 11, and the frames used for peak quantification were 12 to 16.” We made sure that information about the image acquisition frame rate was provided earlier in the text.

- Clarify how the labellum was covered in Figure 3 and show that this does not affect the fly's ability to do PER (e.g., test PER to sugar stimulation on tarsus) - otherwise one might think that gluing the labella could affect PER.

In Figure 3, only calcium responses were recorded, and PER was not recorded simultaneously from the same flies. To ensure stable recording from GRN axons in the SEZ, we kept the fly’s proboscis in an extended position as gently as possible using a strip of parafilm. In some of the imaging experiments, we covered the labellum with UV curable glue, whose purpose was not to fix the labellum in an extended position but to prevent the odors from interacting with GRNs on the labellum. We have added a text in the Methods to explain how we covered the labellum.

- Clarify how the coefficients for the linear equation were chosen in Figure 3G.

We used linear regression (implemented in Python using scikit-learn) to model the relationship between neural activity and behavior, aiming to predict the PER duration based on the calcium responses of two GRN types, Gr5a and Gr66a. The coefficients were estimated using the LinearRegression function. We added this description to the Methods.

- Typo in "L-type", Figure 4A.

We appreciate the reviewer for pointing out this error and have corrected it.

- Clarify over what time period ephys recordings were averaged to obtain average responses.

We have modified the description in the Methods as follows: “The average firing rate was quantified by using the spikes generated between 200 and 700 ms after the stimulus contact following the convention to avoid the contamination of motion artifact (Dahanukar and Benton, 2023; Delventhal et al., 2014; Hiroi et al., 2002).

- The data and statistics indicate that MCH does not enhance feeding in Figure 6G, so the text in lines 207-208 is not accurate.

We have modified the text as follows: “A similar result was observed with ethyl butyrate, and a slight, although not significant, increase was also observed with 4-methylcyclohexanol (Figure 6G).”

- P-value for Figure S9 correlation is not reported.

We appreciate the reviewer for pointing this out. The p-value is 0.00044, and we have added it to the figure legend (current Figure 5-figure supplement 1).

**Reviewer #2 (Recommendations for the authors):**
Honestly, I have no recommendations for improvement. The manuscript is extremely well-written and logical. The experiments are persuasive. A lapidary piece of work.

We appreciate the reviewer for the positive assessment of our work.

**Reviewer #3 (Recommendations for the authors):**
- I suggest explaining the rationale for selecting a 4-second interval, beginning 1 second after the onset of stimulation.

Integrated PER duration was defined as the sum of PER duration over 4 s starting 1 s after the odor onset. This definition was set based on the following data.

(1) We used a photoionization detector (PID) to measure the actual time that the odor reaches the position of a tethered fly, which was approximately 1.1 seconds after the odor valve was opened. Therefore, we began analyzing PER responses 1 second after the odor onset (valve opening) to align with the actual timing of stimulation.

(2) As shown in Fig.1D and 1F, the majority of PER occurred within 4 s after the odor arrival.

We have now added the above rationale in the Methods.

- I could not find the statistical analysis for Figures 1E and 1G. If these figures are descriptive, I suggest the authors revise the sentences: 'Unexpectedly, we found that the odors alone evoked repetitive PER without an application of a tastant (Figures 1D-1G, and Movie S1). Different odors evoked PER with different probability (Figure 1E), latency (Figure S1A), and duration (Figures 1F, 1G, and S2)'.

We have added the results of statistical analysis to the figure legend.

- In Figure 2, the authors performed a Scheirer-Ray-Hare test, which, to my knowledge, is a nonparametric test for comparing responses across more than two groups with two factors. If this is the case, please provide the p-values for both factors and their interaction

We now show the p-values for both factors, odor and group as well as their interaction in the supplementary table.

- In line 83, I suggest the authors avoid claiming that 'these data show the olfactory system modulates but is not required for odor-evoked PER,' as they are inhibiting most, but not all olfactory receptor neurons. In this regard, is it possible to measure the olfactory response to odors in these flies?

We thank the reviewer for the comment. Because Orco-Gal4 does not label all the ORNs and because we have more concrete results on flies in which all the olfactory organs are removed as well as specific GRNs and Gr are manipulated, we decided to remove the data for Orco>kir2.1 flies and have updated the text and Figure 2 accordingly.

- In Figure 2, I wonder if there are differences in the contribution of various receptors in detecting different odors. A more detailed statistical analysis might help address this question.

Although it might be possible to infer the contribution of different gustatory receptors by constructing a quantitative model to predict PER, it is a bit tricky because the activity of individual GRNs and not Grs are manipulated in Figure 2 except for Gr5a. The idea could be tested in the future by more systematically manipulating many Grs that are encoded in the fly genome.

- For Figures 2J-L, please clarify which group serves as the control.

We have added this information to the legend.

- In Figure 3, I recommend including an air control in panels D and F to better appreciate the magnitude of the response under these conditions.

The responses to all three controls, air, mineral oil and water, were almost zero. As the other reviewer suggested to present trial-to-trial variability as well, we now show responses to all the controls in all the trials in all the animals tested in Figure 3-figure supplement 2.

- I had difficulty understanding Figure 3G. Could the authors provide a more detailed explanation of the model?

We used linear regression (implemented in Python using scikit-learn) to model the relationship between neural activity and behavior, aiming to predict the PER duration based on the calcium responses of two GRN types, Gr5a and Gr66a. The weights for GRNs were estimated using the LinearRegression function. The weight for Gr5a and Gr66a was positive and negative, respectively, indicating that Gr5a contributes to enhance whereas Gr66a contributes to reduce PER.

To evaluate the model performance, we calculated the coefficient of determination (R^2^), which was 0.81, meaning the model explained 81% of the variance in the PER data.

The scatter plot in Fig. 3G shows a tight relationship between the predicted PER duration (y-axis) plotted against the actual PER duration (x-axis), demonstrating a strong predictive power of the model.

We added the details to the Methods.

- In Figure S4a, the reported p-value is 0.88, which seems to be a typo, as the text indicates that PER is enhanced in a starved state.

Thank you for pointing this out. We have modified the figure legend to describe that PER was enhanced in a starved state only for the experiments conducted with odors at 10^-1^ concentration (current Figure 2-figure supplement 1).